# UniFast-HGR: Scalable and Efficient Maximal Correlation for Multimodal Models

**Hongkang Zhang** [1]  **Shao-Lun Huang** [1]  **Yanlong Wang** [1]  **Ercan Engin KURUOGLU** [1]

## Abstract

This paper presents UniFast-HGR, a scalable surrogate for Hirschfeld–Gebelein–Rényi (HGR) maximal correlation in high-dimensional multimodal learning. The method replaces explicit covariance whitening with centered and $\ell_2$-normalized cosine alignment, rewrites the covariance interaction in local-batch Gram space, and removes invariant diagonal self-correlation through Trivial Spectrum Suppression (TSS). This reformulation preserves the standardized dependence-maximization target of HGR-style learning while replacing the finite-sample whitening estimator with a differentiable local-batch objective whose dominant structural cost is $O(m^2K)$ for local batch size $m$ and feature dimension $K$. OptFast-HGR further reduces the practical memory burden by estimating the off-diagonal structural term through stochastic projection. Experiments across retrieval, image classification, remote sensing segmentation, and multimodal emotion recognition show consistent gains over covariance-based HGR/CCA variants and contrastive or neural MI-estimator objectives on strong multimodal backbones, while microbenchmarks confirm stable behavior at extreme feature dimensions.

## 1. Introduction

Robust representation learning across modalities remains a central problem in modern machine learning (Bengio et al., 2013). Foundation-model training has largely relied on contrastive objectives—exemplified by InfoNCE (Oord et al., 2018) and CLIP (Radford et al., 2021)—which bring matched samples together and separate mismatched samples in a normalized embedding space. While effective for massive, balanced datasets, these geometry-based objectives can degrade under real-world imperfections. Since they do not directly optimize an HGR-style statistical dependence criterion, they can reflect modality-specific artifacts when data contain high-noise modalities (Robinson et al., 2021) or long-tail distributions. This issue becomes more complex in multimodal data, which combine information from diverse sources such as images, text, and audio (Summaira et al., 2021). Traditional linear methods, such as Canonical Correlation Analysis (CCA) (Hotelling, 1936), and Euclidean distance minimization approaches (Frome et al., 2013), are often insufficient to capture the highly nonlinear interaction structures prevalent in complex neural architectures (Baltrušaitis et al., 2019; Guo et al., 2019; Gandhi et al., 2023).

The Hirschfeld–Gebelein–Rényi (HGR) maximal correlation (Hirschfeld, 1935; Gebelein, 1941; Rényi, 1959) provides a classical theoretical reference point for this goal. Generalizing CCA to nonlinear settings, HGR offers a strong dependence criterion for extracting informative shared structure while reducing modality-specific noise (Huang et al., 2017). However, the transition from theory to practice is obstructed by rigid whitening constraints. HGR requires centered and whitened feature representations, which impose a cubic computational burden ($O(K^3)$) through matrix inversion and decomposition operations. In deep neural networks, these operations are computationally costly and susceptible to numerical instability, limiting scalability. Several efforts have attempted to approximate HGR in deep learning. Extensions such as Kernel CCA (Akaho, 2001) and Deep CCA (Andrew et al., 2013) bridge this gap but still inherit the computational cost of whitening. Alternative relaxations like Soft-CCA (Chang et al., 2018) and Correlational Neural Networks (Chandar et al., 2016) relax some constraints but remain sensitive to covariance estimation and batch statistics. Furthermore, the classical HGR framework is often agnostic to supervised tasks, assuming discriminative information resides solely in the shared subspace. While Maximal Correlation Regression (MCR) (Xu & Huang, 2020) attempted to resolve this by linking HGR to linear discriminant analysis, and related research has explored sample complexity via Alternating Conditional Expectations (ACE) (Huang & Xu, 2021) or incorporated

---

[1]Tsinghua Shenzhen International Graduate School, Tsinghua University, Shenzhen, P.R. China. Correspondence to: Shao-Lun Huang <shaolun.huang@sz.tsinghua.edu.cn>.

*Proceedings of the 43$^{rd}$ International Conference on Machine Learning*, Seoul, South Korea. PMLR 306, 2026. Copyright 2026 by the author(s).

HGR into person recognition loss functions (Liang et al., 2021), these methods struggle to scale to foundation-model dimensions.

The most relevant neural approximation, Soft-HGR (Wang et al., 2019), relaxes exact whitening through a soft covariance regularizer. It has found utility in audio-visual emotion recognition (Ma et al., 2021), data augmentation via multimodal GANs (Ma et al., 2022), and correlation analysis in MultiEMO (Shi & Huang, 2023). Yet, Soft-HGR remains limited: it still relies on explicit covariance interactions that require $O(K^2)$ memory. In the Gram-space view, this covariance interaction also includes fixed diagonal self-similarities that carry no inter-sample information. As model dimensions grow, the covariance bottleneck and the diagonal self-correlation bias can weaken informative off-diagonal dependence and make covariance-based HGR approximations difficult to use in state-of-the-art deep learning architectures.

UniFast-HGR bridges the gap between statistical rigor and computational scalability by reformulating maximal correlation as a lightweight spectral regularizer. The instability in prior approximations is traced to the trivial spectrum: fixed diagonal entries in the Gram structure represent self-correlation, dominate part of the structural term, and mask subtle inter-sample dependence. UniFast-HGR replaces explicit covariance computations with a centered cosine formulation, moves the structural interaction into Gram space, and applies diagonal removal, or *Trivial Spectrum Suppression* (TSS), before evaluating the structural term. The objective is best understood as an HGR → Soft-HGR → Gram-space surrogate: it preserves the standardized dependence-maximization target while replacing the finite-sample whitening estimator.

For massive-batch pretraining, **OptFast-HGR** provides a stochastic approximation to the same off-diagonal structural signal. Because the regularizer acts on latent feature matrices, it can be attached to contrastive, self-supervised, or supervised backbones without changing the encoder architecture. It also keeps the auxiliary loss scale bounded through cosine normalization, which supports stable use under high-dimensional feature maps and small local batches. The contributions are summarized as follows:

**Unified Efficiency and Scalability**: UniFast-HGR provides an HGR-style formulation compatible with modern high-dimensional models. By replacing covariance whitening with an $\ell_2$-normalized cosine formulation, it eliminates matrix decomposition and reduces the dominant cost to $O(m^2K)$ for local batch size $m$ and feature dimension $K$. This changes maximal correlation from a covariance bottleneck into a feasible drop-in auxiliary regularizer.

**Enhanced Discriminative and Correlation Power**: TSS is

formulated as an explicit off-diagonal regularization principle that suppresses invariant self-correlation. Coupled with variance control, it improves robustness under noise, batch imbalance, homogeneous batches, and missing-modality regimes, while aligning discriminative learning with dependence maximization.

**Overcoming Complexity Limitations via OptFast**: OptFast-HGR reduces the practical memory and runtime overhead of the full Gram objective through stochastic projection. It preserves the same structural dependence signal while approaching the efficiency profile of simple dot-product operations in large-batch settings.

## 2. Related Work

The quantification and maximization of dependence between high-dimensional variables is a foundational problem that has evolved from classical statistical methods to modern deep learning estimators.

**Canonical Correlation Analysis and Deep Variants.** Canonical Correlation Analysis (CCA) (Hotelling, 1936) seeks linear projections that maximize the correlation between two views. To capture nonlinear relationships, Kernel CCA (KCCA) (Akaho, 2001) maps data into Reproducing Kernel Hilbert Spaces (RKHS), though it scales poorly due to the prohibitive cost of storing and inverting large sample-level kernel matrices. Deep CCA (DCCA) (Andrew et al., 2013) extends this by parametrizing transformations with neural networks. However, DCCA inherits the *cubic computational bottleneck* of CCA: it requires computing the covariance matrix of network outputs and performing eigendecomposition, an $O(K^3)$ operation. This becomes numerically unstable for wide networks or limited batch sizes. Recent efforts have focused on stabilizing this process. **Stabilized DCCA** (He et al., 2024) introduces noise injection to prevent singular covariance matrices in high-dimensional regimes, while probabilistic CCA variants model shared latent factors explicitly (Karami & Schuurmans, 2021). Despite these algorithmic refinements, the fundamental reliance on covariance estimation or model-specific latent-variable inference persists, imposing a strict scalability barrier. Soft-CCA (Chang et al., 2018) relaxes orthogonality via stochastic decorrelation, but it can still suffer from high gradient variance.

**HGR Maximal Correlation and Neural Approximations.** The HGR maximal correlation (Rényi, 1959) generalizes CCA to arbitrary measurable functions and provides a strong nonlinear dependence criterion. While iterative estimators like ACE (Breiman & Friedman, 1985) exist, they are computationally impractical for deep learning. **Soft-HGR** (Wang et al., 2019) introduced a neural approximation of HGR by relaxing rigid whitening constraints into a soft

covariance regularizer. This removes the cubic whitening step but retains a quadratic covariance interaction. However, Soft-HGR still has a *memory bottleneck*: it requires explicit $K \times K$ feature covariance matrices. In foundation-model regimes where feature dimensions are large ($K \geq 1024$), the $O(K^2)$ memory footprint can trigger out-of-memory (OOM) errors. A recent variant, **I-SoftHGR** (Zhang et al., 2024), improves centering and normalization stability but retains the covariance-based formulation, so the underlying scalability limit remains.

**Scalable and Interpretable Dependence Estimators.** Parallel to correlation maximization, lightweight estimators have been proposed primarily for representation analysis. **Centered Kernel Alignment (CKA)** (Kornblith et al., 2019) measures similarity but requires pairwise sample kernels in common variants, which limits its utility as an online training objective. **Distance Correlation (dCor)** (Zhen et al., 2022) captures nonlinear dependence with $O(N^2)$ complexity but lacks the variance-normalization property of HGR, making it sensitive to scaling artifacts. Most recently, $I_d$**Cor** (Basile et al., 2025) proposes an intrinsic-dimension-aware estimator for representation comparison. While such estimators offer valuable theoretical insights for *evaluating* representations, they are primarily designed for post-hoc analysis rather than as differentiable, discriminative regularizers for large-scale pretraining. Current literature therefore leaves a practical gap: HGR/CCA methods are statistically grounded but computationally heavy, while contrastive objectives are scalable but do not directly optimize HGR-style dependence. Neural mutual-information estimators such as MINE, NWJ, and CLUB introduce variational or contrastive density-ratio critics (Belghazi et al., 2018; Nguyen et al., 2010; Cheng et al., 2020). UniFast-HGR instead keeps the HGR dependence view and removes the covariance bottleneck through a normalized Gram-space objective that can regularize contrastive, self-supervised, or supervised backbones.

# 3. Proposed Method

UniFast-HGR is a scalable reformulation of maximal correlation for high-dimensional multimodal learning. The derivation follows the path from classical HGR to Soft-HGR and then to a Gram-space surrogate: the standardized dependence-maximization target is retained, while exact finite-sample whitening is replaced by centered cosine alignment and an off-diagonal structural term. OptFast-HGR further approximates the same structural signal for large-batch settings.

Figure 1 separates the estimator from the encoder architecture. The classical route regularizes feature dimensions through covariance matrices, so its cost and conditioning are governed by $K$ and by finite-batch covariance estimates.

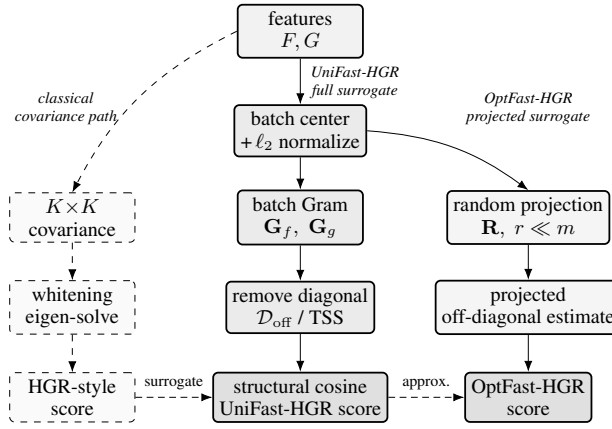

*Figure 1.* Conceptual computation paths. The dashed left branch follows the classical covariance-based HGR pipeline. UniFast-HGR replaces covariance whitening with centered and $\ell_2$-normalized Gram-space dependence, then suppresses invariant diagonal self-correlation through $\mathcal{D}_{\text{off}}$. OptFast-HGR approximates the same off-diagonal structural signal through random projection to reduce memory and large-batch overhead.

UniFast-HGR keeps the paired-sample alignment term but evaluates structural dependence through normalized Gram rows after diagonal suppression. OptFast-HGR changes only the estimator of this off-diagonal relation; the task loss, encoders, and downstream protocols remain unchanged. Throughout complexity statements, $m$ denotes the local per-device batch size used to build the Gram matrix, while $N$ denotes the finite batch size in empirical derivations and sensitivity studies.

## 3.1. Preliminary: HGR and the Covariance Bottleneck

Let $\mathcal{X}$ and $\mathcal{Y}$ be two random variables (modalities) with joint distribution $P_{XY}$. The HGR maximal correlation seeks transformations $f : \mathcal{X} \to \mathbb{R}^K$ and $g : \mathcal{Y} \to \mathbb{R}^K$ that maximize the expected correlation under strict whitening constraints. Given $N$ samples with feature matrices $f \in \mathbb{R}^{N \times K}$ and $g \in \mathbb{R}^{N \times K}$, the objective is:

$$\rho_{\text{HGR}}(X, Y) = \sup_{\substack{f,g \\ \mathbb{E}[f]=\mathbb{E}[g]=0 \\ \text{cov}(f)=\text{cov}(g)=I}} \mathbb{E}\left[f(X)^\top g(Y)\right] \quad (1)$$

where $\text{cov}(f) = \mathbb{E}[ff^\top]$. The constraint $\text{cov}(f) = I$ ensures that the learned features are orthogonal and decorrelated. In practice, enforcing this requires computing the inverse square root of the covariance matrix, an operation with $O(K^3)$ complexity.

Soft-HGR (Wang et al., 2019) relaxes this hard constraint by replacing exact whitening with a soft covariance interaction:

$$\mathcal{J}_{\text{Soft}} = \mathbb{E}[f^\top g] - \tfrac{1}{2}\text{tr}(\text{cov}(f)\,\text{cov}(g)), \quad \mathbb{E}[f] = \mathbb{E}[g] = 0 \quad (2)$$

This objective consists of two inner products: one between

paired feature mappings and one between feature covariance matrices. It avoids explicit whitening, but still requires the construction and storage of covariance matrices $\text{cov}(f) \in \mathbb{R}^{K \times K}$. This formulation becomes memory-prohibitive ($O(K^2)$ memory) and numerically fragile when feature dimension $K$ is large (e.g., $K = 1024$ in modern ViTs). The reformulation below is used as a scalable HGR-style training objective for high-dimensional representations. It preserves dependence maximization under centering and standardized variance constraints, while replacing exact whitening by row normalization and a Gram-space structural term. The alignment direction is retained, and the covariance interaction is estimated through sample relations instead of explicit feature covariance matrices.

### 3.2. Variance-Constrained Reformulation

A critical limitation of covariance-based HGR objectives is their sensitivity to signal variance. Without explicit normalization, feature magnitude can affect both the alignment term and the covariance interaction. To address this, UniFast-HGR applies local-batch variance control followed by row-wise normalization.

Starting from the correlation term in Eq. (1), the expectation is expanded over a finite batch of size $N$. Using the same finite-sample scaling as the covariance estimator, the empirical alignment term is written as

$$\mathbb{E}\big[f(X)^\top g(Y)\big] \approx \frac{1}{N-1} \sum_{i=1}^{N} f(x_i)^\top g(y_i). \quad (3)$$

To enforce the zero-mean constraint, the local batch mean is subtracted from each feature. The centered alignment term becomes

$$\mathbb{E}\big[f^\top g\big] = \frac{1}{N-1} \sum_{i=1}^{N} (f(x_i) - \mathbb{E}[f])^\top (g(y_i) - \mathbb{E}[g]). \quad (4)$$

A standardized variance constraint is then introduced to stabilize the objective. Let $\mu_f, \sigma_f$ and $\mu_g, \sigma_g$ denote local-batch statistics, and let $\epsilon$ be a small numerical constant. The standardized features are

$$\hat{f}_i = \frac{f(x_i) - \mu_f}{\sigma_f + \epsilon}, \qquad \hat{g}_i = \frac{g(y_i) - \mu_g}{\sigma_g + \epsilon}. \quad (5)$$

Row-wise $\ell_2$ normalization is then applied to control sample-wise magnitude:

$$\widetilde{f}_i = \frac{\hat{f}_i}{\|\hat{f}_i\|_2 + \epsilon}, \qquad \widetilde{g}_i = \frac{\hat{g}_i}{\|\hat{g}_i\|_2 + \epsilon}. \quad (6)$$

This normalization bounds the paired alignment term and reduces gradient growth caused by high-dimensional feature magnitudes. Under row-wise normalization, the inner

product becomes cosine similarity:

$$\left\langle \widetilde{f}_i, \widetilde{g}_i \right\rangle = \frac{\hat{f}_i^\top \hat{g}_i}{\|\hat{f}_i\|_2 \|\hat{g}_i\|_2} = \cos(\hat{f}_i, \hat{g}_i). \quad (7)$$

This enables efficient cosine implementations and replaces unstable covariance whitening with normalized geometric alignment in the subsequent Gram-space derivation.

### 3.3. Spectral Expansion in the Gram Space

The covariance interaction in the Soft-HGR objective is the main high-dimensional term. To remove the feature-covariance storage bottleneck, this term is expanded in the sample interaction space, or Gram space. The trace interaction is

$$\text{trace} = \frac{1}{2} \text{tr}(\text{cov}(f) \, \text{cov}(g)). \quad (8)$$

Recall the definition of sample covariance matrices for zero-mean features. Instead of computing them directly, they can be expressed as outer products:

$$\text{cov}[f] = \tfrac{1}{N-1} \sum_{i=1}^{N} (f(x_i) - \mathbb{E}[f]) (f(x_i) - \mathbb{E}[f])^T \quad (9)$$

$$\text{cov}[g] = \tfrac{1}{N-1} \sum_{i=1}^{N} (g(y_i) - \mathbb{E}[g]) (g(y_i) - \mathbb{E}[g])^T \quad (10)$$

Let $\widetilde{\mathbf{F}}, \widetilde{\mathbf{G}} \in \mathbb{R}^{N \times K}$ denote the centered, standardized, and row-normalized feature matrices. Up to the shared covariance normalization factor, the covariance interaction can be written as

$$\text{tr}\left(\widetilde{\mathbf{F}}^\top \widetilde{\mathbf{F}} \, \widetilde{\mathbf{G}}^\top \widetilde{\mathbf{G}}\right) = \text{tr}\left(\widetilde{\mathbf{F}} \widetilde{\mathbf{F}}^\top \widetilde{\mathbf{G}} \widetilde{\mathbf{G}}^\top\right) = \langle \mathbf{G}_f, \mathbf{G}_g \rangle_F. \quad (11)$$

Here

$$\mathbf{G}_f = \widetilde{\mathbf{F}} \widetilde{\mathbf{F}}^\top, \qquad \mathbf{G}_g = \widetilde{\mathbf{G}} \widetilde{\mathbf{G}}^\top. \quad (12)$$

This identity follows from the cyclic property of the trace. It shows that the feature-covariance interaction can be evaluated through sample-sample relations, avoiding explicit $K \times K$ covariance storage while preserving the corresponding structural comparison.

With the variance constraint and row normalization from the previous subsection, each row of the Gram matrix represents a normalized similarity distribution over the local batch. The trace interaction can then be implemented as a correlation of relational structures:

$$\text{trace} \propto \sum_{i=1}^{N} \cos((\mathbf{G}_f)_{i:}, (\mathbf{G}_g)_{i:}). \quad (13)$$

Thus, the trace interaction is estimated by comparing the *relational structures* of the samples. This shifts computation from feature covariance space, which requires $O(K^2)$

storage and costly whitening, to sample interaction space with $O(N^2)$ Gram relations. Define the distribution vector, or Gram matrix row, as

$$\text{distri}_{f,i} = (\mathbf{G}_f)_{i:}, \qquad \text{distri}_{g,i} = (\mathbf{G}_g)_{i:}. \qquad (14)$$

The trace surrogate becomes the cosine similarity between these distribution vectors:

$$\text{trace} \approx \frac{1}{2(N-1)} \sum_{i=1}^{N} \cos(\text{distri}_{f,i}, \text{distri}_{g,i}). \qquad (15)$$

### 3.4. Trivial Spectrum Suppression (TSS)

While Eq. (15) captures the correlation structure, it contains an invariant diagonal component. Since features are projected onto the unit hypersphere after row-wise normalization, the diagonal elements of the Gram matrix are fixed self-similarities: $(\mathbf{G}_f)_{ii} = \langle \widetilde{f}_i, \widetilde{f}_i \rangle = \|\widetilde{f}_i\|_2^2 \approx 1, (\mathbf{G}_g)_{ii} \approx 1$. This phenomenon and its implication are formalized as follows:

**Definition 1 (Trivial Spectrum).** *Let $\mathbf{G} \in \mathbb{R}^{N \times N}$ be the Gram matrix of unit-norm vectors. The Trivial Spectrum is defined as the component of the spectral distribution $\rho(\mathbf{G})$ induced by the diagonal constraints $(\mathbf{G})_{ii} \approx 1$. This component represents static self-correlation energy and does not encode relations between distinct samples.*

In gradient-based optimization, retaining this trivial spectrum consumes part of the structural term with invariant self-similarities. It can reduce the relative weight of off-diagonal entries, especially in small or homogeneous batches. To remove this invariant component, UniFast-HGR applies Trivial Spectrum Suppression (TSS).

The off-diagonal operator $\mathcal{D}_{\text{off}} : \mathbb{R}^{N \times N} \to \mathbb{R}^{N \times N}$ is defined as $\mathcal{D}_{\text{off}}(\mathbf{M}) = \mathbf{M} - \text{Diag}(\text{diag}(\mathbf{M}))$. The structural regularization term is then computed only from the off-diagonal Gram rows:

$$\mathcal{J}_{\text{struct}} = \frac{1}{N-1} \sum_{i=1}^{N} \cos((\mathcal{D}_{\text{off}}(\mathbf{G}_f))_{i:}, (\mathcal{D}_{\text{off}}(\mathbf{G}_g))_{i:}). \qquad (16)$$

By suppressing the trivial spectrum, the structural comparison depends on distinct-sample relations ($i \neq j$) and avoids self-identity artifacts in the Gram rows.

Combining the cosine alignment term with TSS gives the UniFast-HGR objective:

$$\mathcal{J}_{\text{UniFast}} = \frac{1}{N-1} \sum_{i=1}^{N} \cos(\widetilde{f}_i, \widetilde{g}_i)$$
$$- \frac{\lambda}{2(N-1)} \sum_{i=1}^{N} \cos((\mathcal{D}_{\text{off}}(\mathbf{G}_f))_{i:}, (\mathcal{D}_{\text{off}}(\mathbf{G}_g))_{i:}). \qquad (17)$$

Operationally, Eq. (17) is implemented as a standard differentiable auxiliary loss and is added to the task-specific objective. The coefficient $\lambda$ controls the strength of the off-diagonal structural term. Because both the alignment and structural terms are bounded by normalization, the same scale works across tasks; $\lambda = 0.5$ is the default except for IEMOCAP, where $\lambda = 1.0$ is used because the supervised loss is stronger relative to the dependence term. Two implementation details are used to keep this surrogate well conditioned. Batch centering is applied before normalization, and a small numerical constant is added to each norm to avoid unstable gradients in nearly constant features. The off-diagonal operator is applied before computing the row-wise structural cosine, so the row direction is determined only by cross-sample relations. Thus the diagonal mass never contributes to either the numerator or denominator of the structural term. This detail is important for small or imbalanced batches, where the invariant diagonal can otherwise dominate the scale of the Gram rows. Geometrically, suppressing invariant diagonal mass and emphasizing off-diagonal relations is consistent with a more uniform angular arrangement of feature vectors on the hypersphere. This supports separability in contrastive and discriminative learning settings without requiring explicit ETF constraints.

### 3.5. Multimodal Generalization

The HGR maximal correlation was originally defined for two variables. Extending it to multiple modalities introduces new whitening constraints that traditionally increase computational complexity. UniFast-HGR offers enhanced flexibility here. For a set of $M$ modalities $\{X^{(1)}, \ldots, X^{(M)}\}$, the multimodal objective aggregates pairwise maximal correlations:

$$\mathcal{J}_{\text{Multi}} = \sum_{1 \leq u < v \leq M} \left[ \frac{1}{N-1} \sum_{i=1}^{N} \cos\left( f_i^{(u)}, f_i^{(v)} \right) \right.$$
$$\left. - \frac{\lambda}{2(N-1)} \sum_{i=1}^{N} \cos\left( \mathcal{D}_{\text{off}}(\text{distri}_{f,i}^{(u)}), \mathcal{D}_{\text{off}}(\text{distri}_{f,i}^{(v)}) \right) \right]. \qquad (18)$$

This formulation is a pairwise aggregation framework rather than a closed-form theory for higher-order multimodal dependence. Uniform pair weights are used as a parameter-free default to isolate the HGR reformulation itself, while redundancy-aware or task-guided pair weights can be introduced without changing the objective. By operating in the Gram space, the computational cost grows linearly with the number of pairs, rather than cubically with the total feature dimension. The unweighted objective is used in the main protocol. A validation-weighted variant gives mild gains on MSR-VTT (R@1: 43.3 to 43.8) and IEMOCAP (F1/ACC: 73.57/73.66 to 73.89/74.02), indicating compatibility with pair weighting.

## 3.6. Speed Optimization: OptFast-HGR

For extreme-scale scenarios with large local feature blocks, the $O(m^2)$ pairwise interaction term can become a bottleneck. OptFast-HGR estimates the off-diagonal structural term by random projection. Instead of comparing the full $m \times m$ Gram structure, it projects the off-diagonal signal into a lower-dimensional sketch. Given a normalized local feature matrix $\mathbf{F} \in \mathbb{R}^{m \times K}$ and a random projection matrix $\mathbf{R} \in \mathbb{R}^{m \times r}$ with $r \ll m$, the projected structure is

$$\widehat{\mathbf{G}}_f^{\mathrm{proj}} = \mathcal{D}_{\mathrm{off}}(\mathbf{F}\mathbf{F}^\top)\mathbf{R} = \mathbf{F}(\mathbf{F}^\top \mathbf{R}) - \mathrm{Diag}(\mathbf{d}_F)\mathbf{R},$$
$$\mathbf{d}_F = \mathrm{diag}(\mathbf{F}\mathbf{F}^\top). \tag{19}$$

For row-normalized features, $\mathbf{d}_F \approx \mathbf{1}$, so the diagonal correction is cheap and does not require storing the full Gram matrix. The same projection is applied to the paired modality before computing the structural cosine. This reduces the dominant structural computation to $O(mKr)$ and the sketch memory to $O(mr)$, approaching $O(mK)$ when $r$ is fixed and small. The approximation error is controlled by the projection dimension and the off-diagonal energy, as discussed in Appendix B. Algorithm 2 in Appendix A.5 gives the implementation.

## 3.7. Complexity Analysis and End-to-End Scalability

The practical integration of UniFast-HGR into foundation model pipelines is governed by computation, peak memory, and numerical stability in high-dimensional regimes ($K \gg 1024$). Full UniFast-HGR avoids covariance whitening and has $O(m^2K)$ computation for local batch size $m$ and feature dimension $K$. Here $m$ denotes the local per-device batch size used to form the Gram operator, not the global batch size after distributed aggregation or gradient accumulation. This distinction matters in foundation-model training: the global batch may be very large, while the local feature block remains moderate and fits the pairwise similarity regime already used by contrastive objectives. It also makes the memory accounting compatible with data-parallel training, since the Gram operator is formed before cross-device aggregation. The main Gram storage scales with $m^2$ rather than $K^2$. Full UniFast-HGR therefore has a quadratic dependence on the local batch, whereas OptFast-HGR reduces the effective memory pressure when the local block must be enlarged.

**Scalability to Extreme Dimensions** ($K = 10^5$): Standard correlation methods face severe bottlenecks as feature dimensions grow. **CCA** relies on matrix inversion with cubic complexity $O(K^3)$, making it computationally intractable beyond $K = 10^3$. **Soft-HGR** requires storing and optimizing $K \times K$ covariance matrices ($O(K^2)$ memory). As shown in Table 1, at $K = 10^4$, Soft-HGR consumes 15.3 GB of VRAM, approaching the limit of standard GPUs. At $K = 10^5$, it fails due to out-of-memory (OOM) errors.

**Comparison with Modern SSL Baselines**: The analysis also includes modern self-supervised learning (SSL) objectives—**VICReg** (Bardes et al., 2022), **Barlow Twins** (Zbontar et al., 2021), and **InfoNCE** (Oord et al., 2018)—as reference points. These methods do not optimize HGR maximal correlation; they are included as optimized efficiency references under the same fixed-local-batch protocol. UniFast-HGR performs the structural comparison in local-batch Gram space. Its Gram operator depends on $m$, while feature storage remains linear in $K$. As reported in Table 1, UniFast-HGR runs stably at $K = 10^5$ with **1.55 GB** of VRAM. This result shows that the covariance whitening bottleneck has been removed while keeping an efficiency profile close to the lightweight SSL references.

*Table 1.* **Scalability Microbenchmark on Extreme Feature Dimensions.** Runtime and peak memory are measured on synthetic features with local batch size $m = 256$ using a single 24GB GPU. **OOM**: out of memory ($> 24$GB); **OOT**: out of time ($> 300$s). **Reference Points**: InfoNCE, VICReg, and Barlow Twins are included as optimized efficiency references rather than HGR/CCA methods. *Note*: Standard implementations of VICReg and Barlow Twins may become impractical at $K = 10^5$ due to $K \times K$ covariance or cross-correlation matrices; the reported reference values assume optimized linear-memory implementations or gradient checkpointing. UniFast-HGR stays in the same efficiency range as these lightweight reference objectives while avoiding the covariance whitening and eigensolve bottlenecks of traditional HGR/CCA methods.

| METHOD | FEATURE DIM. ($K$) | TIME (S) $\downarrow$ | VRAM (GB) $\downarrow$ | STATUS |
|---|---|---|---|---|
| *HGR-Based Methods (Proposed & Baselines)* | | | | |
| SOFT-HGR | $10^3$ | 1.80 | 0.98 | STABLE |
| | $10^4$ | 95.6 | 15.3 | HIGH COST |
| | $10^5$ | – | $> 24.0$ | **OOM** |
| CCA | $10^3$ | 12.4 | 2.50 | SLOW |
| | $10^4$ | – | $> 24.0$ | **OOM** |
| **UNIFAST-HGR** | $10^3$ | **0.07** | **0.82** | **STABLE** |
| | $10^4$ | **0.71** | **1.12** | **STABLE** |
| | $10^5$ | **7.20** | **1.55** | **STABLE** |
| *Efficiency References (Non-HGR Objectives)* | | | | |
| INFONCE | $10^5$ | 6.85 | 1.45 | REFERENCE |
| VICREG | $10^5$ | 7.15 | 1.62 | REFERENCE |
| BARLOW TWINS | $10^5$ | 7.28 | 1.68 | REFERENCE |

## 4. Experiments

The experiments evaluate UniFast-HGR across several scales and tasks: foundation-model tuning on ImageNet, COCO, and InternVid-style video-text retrieval; robustness tests on IEMOCAP; and scientific remote sensing benchmarks. The comparisons include standard contrastive losses, nonlinear dependence estimators such as CKA, dCor, and $I_d$Cor (Basile et al., 2025), covariance-based HGR/CCA variants, and lightweight similarity objectives. The evaluation focuses on three claims: (1) **Scalability**: UniFast-HGR can be integrated into modern ViT/CLIP-style architectures; (2) **Accuracy**: it improves over geometric and correlation-based objectives under matched protocols; and (3) **Robustness**: it reduces degradation under noisy, missing-modality, and limited-label settings.

*Table 2.* **Main Results on Large-Scale Datasets.** UniFast-HGR improves over baselines and recent correlation objectives (dCor, $I_d$Cor) across classification and retrieval tasks.

| Dataset | Model | Baseline | CKA | dCor | $I_d$Cor | Soft-HGR | UniFast-HGR | OptFast-HGR |
|---|---|---|---|---|---|---|---|---|
| | ViT-B/32 | 76.6 | 76.7 | 76.9 | 78.7 | 76.3 | **80.1** | 79.6 |
| ImageNet-1K | ResNet50 | 74.3 | 74.5 | 75.0 | 77.4 | 74.1 | **78.5** | 78.1 |
| Top-1 Accuracy | CLIP | 76.1 | 76.6 | 77.3 | 79.5 | 76.3 | **80.4** | 79.8 |
| (%) | SigLIP | 81.3 | 81.7 | 82.2 | 84.1 | 81.4 | **84.8** | 84.5 |
| | DINOv2 | 81.8 | 82.1 | 82.4 | 84.7 | 81.6 | **85.3** | 84.9 |
| | ViT-B/32 | 38.2 | 38.7 | 39.2 | 39.6 | 38.9 | **40.1** | 39.8 |
| COCO Text-Image | ResNet50 | 37.8 | 38.3 | 38.7 | 39.2 | 38.6 | **39.5** | 39.3 |
| Retrieval Recall@1 | CLIP | 38.9 | 39.5 | 41.4 | 41.7 | 40.3 | **42.1** | 42.0 |
| | SigLIP | 50.8 | 51.3 | 52.8 | 53.2 | 51.6 | **53.8** | 53.5 |
| | DINOv2 | 51.1 | 51.5 | 52.7 | 53.5 | 52.1 | **53.9** | 53.7 |
| *InternVid Video-Text Retrieval (T2V R@1)* | | | | | | | | |
| MSR-VTT | ViCLIP | 36.4 | 37.1 | 37.9 | 38.5 | 38.8 | **43.3** | 42.7 |
| LSMDC | ViCLIP | 17.1 | 17.6 | 18.1 | 18.9 | 18.3 | **20.7** | 20.3 |
| DiDeMo | ViCLIP | 16.4 | 16.9 | 17.3 | 17.8 | 17.6 | **20.5** | 20.1 |

## 4.1. Large-Scale Foundation Model Tuning

The first evaluation validates the method on standard vision-language benchmarks. UniFast-HGR is integrated as an auxiliary regularizer into representative strong backbones, including CLIP (ViT-B/32) (Radford et al., 2021), SigLIP (Zhai et al., 2023), and DINOv2 (Oquab et al., 2024). Comparison baselines include nonlinear correlation and representation-similarity methods such as CKA (Kornblith et al., 2019), dCor (Zhen et al., 2022), and $I_d$Cor (Basile et al., 2025).

**ImageNet-1K Classification**: As shown in Table 2, UniFast-HGR improves performance across architectures. On **DINOv2**, it achieves **85.3%** Top-1 accuracy, giving a **+3.5** percentage-point gain over the baseline. The improvement over $I_d$Cor and Soft-HGR indicates that the Gram-space HGR-style objective provides useful dependence regularization beyond recent correlation measures and covariance-based HGR variants.

**Cross-Modal Retrieval**: On **COCO** (Lin et al., 2014) text-image retrieval, CLIP+UniFast-HGR achieves **42.1% Recall@1**, surpassing baseline CLIP (38.9%) and Soft-HGR (40.3%). OptFast-HGR maintains similar performance (42.0% R@1) while reducing the structural computation, which supports its efficiency-accuracy trade-off.

**Large-Scale Video-Text Retrieval**: The evaluation further extends to video-text retrieval under the **InternVid-10M** setting (Wang et al., 2024). As shown in Table 2, integrating UniFast-HGR into **ViCLIP** gives the highest R@1 on MSR-VTT (Xu et al., 2016), LSMDC (Yao et al., 2015), and DiDeMo (Hendricks et al., 2017), with gains of 6.9, 3.6, and 4.1 R@1 points, respectively, over the baseline. This result shows stable transfer to video-text retrieval, where classical whitening-based HGR/CCA objectives are difficult to apply at scale.

## 4.2. Image Classification

The performance of UniFast-HGR was evaluated against CCA, Deep CCA, Soft CCA, Soft-HGR, cosine similarity, and dot product in image classification. Comparative experiments were conducted using a dual-channel deep learning framework for remote sensing data classification, with ResNet-50 (He et al., 2016) as the backbone. Following the same conditions and preprocessing steps as Wu et al. (Wu et al., 2022), classification results on the Berlin dataset (Hong et al., 2021; Okujeni et al., 2016) are presented in Table 3. Performance is evaluated using overall accuracy (OA), average accuracy (AA), and kappa coefficient. On the Berlin and Houston 2018 (Lin et al., 2023) datasets, UniFast-HGR improves classification accuracy. OptFast-HGR uses the projected off-diagonal structural estimate and achieves computational efficiency close to dot-product and cosine-similarity operations. These results show that UniFast-HGR and OptFast-HGR improve classification performance while keeping moderate runtime. Detailed Berlin/Houston results are reported in Appendix D.1; additional CIFAR-100 results are reported in Appendix D.4.

*Table 3.* Image classification results on the Berlin dataset.

| Methods | OA(%) | AA(%) | Kappa (%) | Time (s/epoch) |
|---|---|---|---|---|
| CCA | 70.93 | 64.35 | 58.28 | 2967.52 |
| Deep CCA | 72.74 | 65.08 | 60.23 | 250.51 |
| Soft CCA | 71.54 | 61.14 | 58.33 | 314.93 |
| Dot Product | 75.20 | 66.22 | 62.77 | **23.18** |
| Cosine Similarity | 75.51 | 65.53 | 62.53 | 23.40 |
| Soft-HGR | 65.80 | 64.30 | 52.99 | 25.83 |
| UniFast-HGR | **80.75** | **71.53** | **70.44** | 24.53 |
| OptFast-HGR | 80.46 | 71.51 | 70.21 | 23.54 |

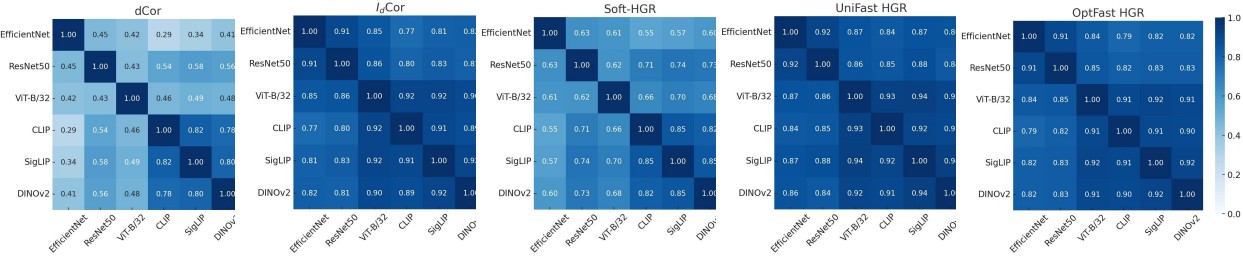

*Figure 2.* Correlation results on ImageNet representations. Detailed values are reported in Appendix D.5.

## 4.3. Remote Sensing Semantic Segmentation

To further evaluate UniFast-HGR and OptFast-HGR, remote sensing semantic segmentation experiments were conducted on the Vaihingen dataset and the large-scale high-resolution annotation dataset Globe230k. The ISPRS Vaihingen dataset, provided by the International Society for Photogrammetry and Remote Sensing (ISPRS) (Wang et al., 2022), is a 2D semantic segmentation dataset with a 9-cm spatial resolution. It includes 8-bit TIFF files for near-infrared, red, and green bands, and a single-band digital surface model (DSM) with 32-bit floating-point height values. The Globe230k dataset contains 232,819 annotated images of size $512 \times 512$ and 1-m spatial resolution, with multiple bands such as RGB and digital elevation models (DEM) (Shi et al., 2023). Using the model and preprocessing steps of Ma et al. (Ma et al., 2024), UniFast-HGR and OptFast-HGR were applied to fuse multimodal remote sensing data. The results are shown in Table 4, evaluated using overall accuracy (OA) and mean intersection over union (mIoU). UniFast-HGR and OptFast-HGR both improve segmentation performance by capturing cross-modal dependence. Detailed experimental results and visualization examples are provided in Appendix D.2.

*Table 4.* Experimental results of remote sensing segmentation.

| Methods | Vaihingen | | Globe230k | |
|---|---|---|---|---|
| | OA(%) | mIoU(%) | OA(%) | mIoU(%) |
| CCA | 91.15 | 79.37 | 87.92 | 67.49 |
| Deep CCA | 91.39 | 81.35 | 88.27 | 67.85 |
| Soft CCA | 91.41 | 81.44 | 87.60 | 66.71 |
| Dot Product | 92.61 | 83.65 | 90.92 | 75.67 |
| Cosine Similarity | 92.56 | 83.34 | 90.81 | 75.53 |
| Soft-HGR | 90.10 | 76.87 | 86.46 | 64.82 |
| UniFast-HGR | **93.01** | **84.62** | **91.48** | **76.36** |
| OptFast-HGR | 92.95 | 84.57 | 91.23 | 76.15 |

## 4.4. Extension to More/Missing Modalities: Multimodal Emotion Recognition

The performance of UniFast-HGR and OptFast-HGR was evaluated in multimodal emotion recognition on the IEMO-CAP dataset. Comparative experiments were conducted using the MultiEMO model proposed by Shi & Huang (Shi & Huang, 2023). Table 5 reports accuracy (ACC) on IEMO-CAP (Busso et al., 2008); detailed W-F1 and class-wise

results are provided in the appendix. Both UniFast-HGR and OptFast-HGR improve performance by capturing cross-modal emotion cues under complete and incomplete inputs. UniFast-HGR was tested under two challenging scenarios: single-modality absence and insufficient labels. In the first scenario, one of the three modalities was excluded, yielding the Text+Audio, Text+Visual, and Audio+Visual settings. In the second scenario, 20%, 50%, or 80% of training labels were hidden. The pairwise aggregation and normalized structural terms reduce sensitivity to missing or noisy modality relations, as shown in Table 5.

## 4.5. Correlation Estimation

To quantify intrinsic alignment capability, cross-model feature correlations were measured on ImageNet embeddings using six representative encoders. This evaluates representation relationships independent of final task heads. Pairwise correlation matrices are computed for Efficient-Net, ResNet50, ViT-B/32, CLIP, SigLIP, and DINOv2 embeddings across 30K random samples. Figure 2 shows that UniFast-HGR yields stronger cross-model correlations. For example, the correlation between CLIP and DINOv2 reaches 0.91 with UniFast-HGR, outperforming dCor (0.78) and Soft-HGR (0.82). UniFast-HGR also gives high correlations for ViT-based models ($\geq 0.92$ in the reported pairs), matching the downstream gains in Table 2. These results indicate that the proposed objective preserves useful feature relationships while improving task performance.

## 4.6. Computational Efficiency

To isolate computational costs from network architecture effects, the benchmark compares correlation calculations between randomly generated tensors. The comparison includes UniFast-HGR and OptFast-HGR against baseline methods across feature dimensions and local batch sizes. For each configuration, paired tensors $f, g \in \mathbb{R}^{m \times K}$ are generated, and the average execution time is measured over 10,000 trials. The experimental results are shown in Figure 4 in Appendix G. The results show that execution time grows as local batch size increases. UniFast-HGR and OptFast-HGR keep low runtime across different local batch sizes, with stronger efficiency advantages at higher feature dimensions. These findings suggest that UniFast-HGR and

*Table 5.* Multimodal emotion recognition results on IEMOCAP (ACC %).

| Methods | No Missing | Missing Modalities | | | Missing Labels | | |
|---|---|---|---|---|---|---|---|
| | Text+Audio+Visual | Text+Audio | Text+Visual | Audio+Visual | 20% | 50% | 80% |
| CCA | 67.41 | 64.55 | 64.03 | 50.71 | 66.21 | 61.63 | 51.91 |
| Deep CCA | 67.78 | 64.92 | 64.38 | 51.06 | 66.50 | 63.10 | 54.80 |
| Soft CCA | 68.58 | 65.68 | 65.27 | 51.89 | 67.35 | 63.81 | 55.43 |
| Dot Product | 70.14 | 67.32 | 67.08 | 53.56 | 69.06 | 65.27 | 57.92 |
| Cosine Similarity | 69.50 | 66.64 | 66.21 | 52.92 | 68.43 | 64.94 | 57.63 |
| Soft-HGR | 71.29 | 67.85 | 67.52 | 53.90 | 69.47 | 65.19 | 57.75 |
| UniFast-HGR | **73.66** | **70.94** | **70.41** | **57.82** | **72.65** | **69.26** | **62.05** |
| OptFast-HGR | 73.43 | 70.67 | 70.15 | 56.57 | 72.39 | 68.92 | 61.58 |

OptFast-HGR capture complex correlations between multimodal data while maintaining practical computational efficiency, making them suitable for multimodal data fusion tasks.

### 4.7. Mechanism Analysis and Sensitivity

Table 6 decomposes the main design choices on Berlin and ImageNet-1K. The cosine reformulation gives the largest gain in this progression. Variance control limits magnitude inflation. The structure-level term adds relational regularization. TSS gives the final improvement by suppressing invariant self-correlation. The OptFast approximation retains most of the gain with lower computational cost.

The structural term uses local-batch similarities, so batch statistics form a natural boundary condition. When the ImageNet local batch size decreases from $m = 256$ to $m = 16$, Soft-HGR drops from 76.3 to 72.1 Top-1, while UniFast-HGR with TSS drops from 80.1 to 78.2. On Berlin, skewed or homogeneous batches also affect Gram-based estimators, but TSS reduces the degradation. Under 80% single-class imbalance, Soft-HGR drops by 6.60 OA points, whereas UniFast-HGR drops by 1.85 points. Under single-label homogeneous batches, the drops are 13.40 and 3.55 points. These results do not imply invariance to arbitrary sampler pathologies. They show that diagonal removal blocks the easiest self-correlation collapse mode and makes the remaining error depend on informative off-diagonal diversity. Appendix F gives the full batch-size, batch-composition, $\lambda$-sensitivity, low-dimensional, pair-weighting, and MI-estimator comparisons.

### 5. Limitations and Future Work

UniFast-HGR and OptFast-HGR target the high-dimensional multimodal regime where covariance-based HGR and CCA variants are costly or unstable. Several boundary cases remain. First, variance constraints can over-regularize very low-dimensional features, where strict normalization may restrict useful task-specific variation. Second, OptFast-HGR may introduce approximation

*Table 6.* Component-wise ablation of the UniFast-HGR design.

| Model Variant | Berlin OA (%) | ImageNet-1K Top-1 (%) |
|---|---|---|
| Baseline (Soft-HGR) | 65.80 | 76.3 |
| + Cosine reformulation | 75.23 | 77.8 |
| + Variance constraints | 78.65 | 79.2 |
| + Structure-level term, diagonal retained | 80.62 | 79.8 |
| + TSS (UniFast-HGR) | **80.75** | **80.1** |
| + OptFast approximation | 80.46 | 79.6 |

bias when paired modalities have strong distributional asymmetry, such as severe long-tail imbalance or highly unequal support. Third, the full Gram-space objective scales quadratically in the local batch size, although OptFast-HGR reduces this pressure through stochastic projection. Finally, diagonal suppression improves robustness to self-correlation bias, but it cannot fully compensate for local batches with little off-diagonal diversity, such as extremely homogeneous or duplicated batches.

These limitations motivate adaptive spectral regularization that adjusts normalization strength according to intrinsic dimensionality, distribution-aware calibration for asymmetric modality pairs, and extensions of diagonal suppression to broader dependency classes, including non-Gaussian, heavy-tailed, and strongly imbalanced settings.

### 6. Conclusion

**UniFast-HGR** provides a scalable surrogate for Hirschfeld–Gebelein–Rényi maximal correlation in high-dimensional multimodal learning. By replacing covariance whitening with variance-controlled cosine alignment, expressing structural dependence in local-batch Gram space, and suppressing invariant diagonal self-correlation through TSS, the objective keeps the HGR-style dependence-maximization view while avoiding the main covariance bottleneck. OptFast-HGR further reduces large-local-batch overhead through a projected estimate of the off-diagonal structural term. Across classification, retrieval, segmentation, and multimodal emotion recognition, the resulting objectives improve over covariance-based HGR/CCA variants and strengthen contemporary multimodal backbones while maintaining a lighter memory and runtime profile.

## Acknowledgements

This work was supported in part by the National Key R&D Program of China under Grant 2021 YFA0715202, Shenzhen Key Laboratory of Ubiquitous Data Enabling (Grant No. ZDSYS20220527171406015) and the Shenzhen Science and Technology Program under Grant KQTD20170810150821146 and Grant JCYJ20220530143002005.

## Impact Statement

This work advances efficient dependence learning for multimodal models. The main positive impact is lower computational cost for training and adapting multimodal systems, which can reduce energy use and broaden access to correlation-based objectives. The same efficiency can also accelerate deployment of multimodal models in high-stakes settings. Practical use should therefore include bias auditing, robustness testing under missing or corrupted modalities, privacy review for multimodal data, and monitoring for failure modes caused by modality imbalance or spurious correlations. The method is an auxiliary objective and does not by itself determine whether a deployed system is safe or fair.

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

# A. Detailed Derivation and Theoretical Grounding

This section derives the UniFast-HGR objective from the Soft-HGR form and connects the Gram-space reformulation to Trivial Spectrum Suppression (TSS) introduced in the main text. In this appendix, $N$ denotes the finite local batch size used in the derivation, corresponding to $m$ in the main text.

## A.1. Step 1: Variance Constraints and Spherical Projection

Soft-HGR relaxes the whitening constraints of classical HGR through a soft covariance interaction:

$$J_{\text{soft}}(f, g) = \mathbb{E}\big[f(X)^T g(Y)\big] - \frac{1}{2} \operatorname{tr}(\operatorname{cov}(f(X)) \operatorname{cov}(g(Y))), \quad \text{s.t.} \quad \mathbb{E}[f] = \mathbb{E}[g] = 0. \tag{20}$$

In the canonical HGR formulation, strict whitening is required. UniFast-HGR keeps the standardized dependence-maximization view but avoids explicit matrix decomposition. It first standardizes local-batch features and then applies row-wise $\ell_2$ normalization. For feature matrices $\mathbf{F}, \mathbf{G} \in \mathbb{R}^{N \times K}$:

$$\widetilde{\mathbf{F}} = \operatorname{Norm}_2\left(\frac{\mathbf{F} - \mu_F}{\sigma_F + \epsilon}\right), \qquad \widetilde{\mathbf{G}} = \operatorname{Norm}_2\left(\frac{\mathbf{G} - \mu_G}{\sigma_G + \epsilon}\right), \tag{21}$$

where $\mu_F, \sigma_F$ and $\mu_G, \sigma_G$ are local-batch statistics, $\epsilon$ is a small numerical constant, and $\operatorname{Norm}_2$ applies row-wise $\ell_2$ normalization. The last operation projects each sample feature onto the unit hypersphere $\mathbb{S}^{K-1}$. Thus, for every sample $i$, $\|\widetilde{\mathbf{F}}_i\|_2 \approx 1$ and $\|\widetilde{\mathbf{G}}_i\|_2 \approx 1$.

## A.2. Step 2: Cosine Reformulation of the Alignment Term

Under row-wise normalization, the inner product becomes a cosine similarity. The first term of the objective is estimated over the finite local batch as follows:

$$\begin{aligned}
\mathbb{E}\big[f(X)^T g(Y)\big] &\approx \frac{1}{N-1} \sum_{i=1}^{N} \left\langle \widetilde{\mathbf{F}}_i, \widetilde{\mathbf{G}}_i \right\rangle \\
&= \frac{1}{N-1} \sum_{i=1}^{N} \|\widetilde{\mathbf{F}}_i\|_2 \|\widetilde{\mathbf{G}}_i\|_2 \cos(\widetilde{\mathbf{F}}_i, \widetilde{\mathbf{G}}_i) \\
&\approx \frac{1}{N-1} \sum_{i=1}^{N} \cos(\widetilde{\mathbf{F}}_i, \widetilde{\mathbf{G}}_i).
\end{aligned} \tag{22}$$

This substitution replaces unbounded dot products with normalized cosine similarities and reduces gradient growth in high-dimensional spaces.

## A.3. Step 3: Trace Term Expansion and Trivial Spectrum Identification

The covariance interaction $\operatorname{tr}(\operatorname{cov}(f) \operatorname{cov}(g))$ captures structural alignment between modalities. This term can be written in Gram space, where rows and columns index samples in the local batch. This algebraic step makes the structural surrogate independent of explicit $K \times K$ covariance storage.

Let $\widetilde{\mathbf{F}}, \widetilde{\mathbf{G}} \in \mathbb{R}^{N \times K}$ be the standardized and row-normalized feature matrices. Up to the shared covariance normalization constant, the covariance matrices are proportional to $\widetilde{\mathbf{F}}^T \widetilde{\mathbf{F}}$ and $\widetilde{\mathbf{G}}^T \widetilde{\mathbf{G}}$. Using the cyclic property of the trace,

$$\operatorname{tr}(\operatorname{cov}(f) \operatorname{cov}(g)) \propto \operatorname{tr}\left((\widetilde{\mathbf{F}}^T \widetilde{\mathbf{F}})(\widetilde{\mathbf{G}}^T \widetilde{\mathbf{G}})\right) \operatorname{tr}\left((\widetilde{\mathbf{F}} \widetilde{\mathbf{F}}^T)(\widetilde{\mathbf{G}} \widetilde{\mathbf{G}}^T)\right) \langle \mathbf{G}_f, \mathbf{G}_g \rangle_F, \tag{23}$$

where

$$\mathbf{G}_f = \widetilde{\mathbf{F}} \widetilde{\mathbf{F}}^T \in \mathbb{R}^{N \times N}, \qquad \mathbf{G}_g = \widetilde{\mathbf{G}} \widetilde{\mathbf{G}}^T \in \mathbb{R}^{N \times N}. \tag{24}$$

These Gram matrices represent pairwise similarities in the local batch and provide the sample-space structural view used by UniFast-HGR.

**The Trivial Spectrum**: Due to the row-wise spherical projection in Step 1, the diagonal elements of the Gram matrix are invariant:

$$(\mathbf{G}_f)_{ii} = \left\langle \widetilde{\mathbf{F}}_i, \widetilde{\mathbf{F}}_i \right\rangle = \|\widetilde{\mathbf{F}}_i\|_2^2 \approx 1, \qquad (\mathbf{G}_g)_{ii} \approx 1. \tag{25}$$

These fixed diagonal entries constitute the **Trivial Spectrum**. They represent fixed self-correlation mass that contributes to the structural term but does not encode relations between distinct samples. Retaining them weakens the relative weight of informative off-diagonal interactions, especially when the local batch is small or homogeneous.

### A.4. Step 4: Trivial Spectrum Suppression (TSS)

The derivation applies the **Trivial Spectrum Suppression (TSS)** operator:

$$\mathcal{D}_{\text{off}}(\mathbf{M}) = \mathbf{M} - \text{Diag}(\text{diag}(\mathbf{M})). \tag{26}$$

Substituting this operation into the Gram-space structural term gives a correlation between off-diagonal sample relations:

$$\mathcal{J}_{\text{struct}} = \frac{1}{N-1} \sum_{i=1}^{N} \cos\big((\mathcal{D}_{\text{off}}(\mathbf{G}_f))_{i:}, (\mathcal{D}_{\text{off}}(\mathbf{G}_g))_{i:}\big). \tag{27}$$

This yields the structural term of UniFast-HGR, which focuses the objective on relations between distinct samples. The complete objective is

$$\mathcal{J}_{\text{UniFast}} = \frac{1}{N-1} \sum_{i=1}^{N} \cos(\widetilde{\mathbf{F}}_i, \widetilde{\mathbf{G}}_i) - \frac{\lambda}{2} \mathcal{J}_{\text{struct}}. \tag{28}$$

When used as a minimization loss, the sign is chosen according to the task-specific training convention.

### A.5. Algorithm Implementation

The procedural steps are provided for UniFast-HGR (Algorithm 1) and its projected variant OptFast-HGR (Algorithm 2). The "Diagonal Removal" step corresponds to the TSS operator derived above.

---

**Algorithm 1** UniFast-HGR Algorithm

---

1: **Input:** Feature matrices $f, g \in \mathbb{R}^{N \times K}$ (batch size $N$, dimension $K$), regularization weight $\lambda$.
2: **Output:** UniFast-HGR objective $\mathcal{J}_{\text{UniFast}}$.

3: *// Step 1: Standardization and row-wise normalization*
4: $f \leftarrow \frac{f - \text{mean}(f)}{\text{std}(f) + \epsilon}, g \leftarrow \frac{g - \text{mean}(g)}{\text{std}(g) + \epsilon}$ {Standardization}
5: $f \leftarrow \text{Norm}_2(f), g \leftarrow \text{Norm}_2(g)$ {Normalize rows}

6: *// Step 2: Alignment term*
7: $\mathcal{J}_{\text{align}} = \frac{1}{N} \sum_{i=1}^{N} \cos(f_i, g_i)$

8: *// Step 3: Gram-space structural term*
9: Compute Gram matrices: $\mathbf{G}_f = f f^\top, \mathbf{G}_g = g g^\top$
10: **Apply TSS**:
11: $\widehat{\mathbf{G}}_f \leftarrow \text{triu}(\mathbf{G}_f, k=1) + \text{tril}(\mathbf{G}_f, k=-1)$ {Zero out diagonal}
12: $\widehat{\mathbf{G}}_g \leftarrow \text{triu}(\mathbf{G}_g, k=1) + \text{tril}(\mathbf{G}_g, k=-1)$

13: *// Step 4: Correlation of off-diagonal structures*
14: $\mathcal{J}_{\text{struct}} = \frac{1}{N} \sum_{i=1}^{N} \text{CosineSimilarity}((\widehat{\mathbf{G}}_f)_{i:}, (\widehat{\mathbf{G}}_g)_{i:})$
15: **return** $\mathcal{J}_{\text{UniFast}} = \mathcal{J}_{\text{align}} - \frac{\lambda}{2} \mathcal{J}_{\text{struct}}$

---

## B. Approximation Analysis of OptFast-HGR

OptFast-HGR accelerates UniFast-HGR by estimating the off-diagonal structural term in a projected subspace. It uses the same alignment term and normalization steps as UniFast-HGR, but replaces the full off-diagonal Gram comparison with a lower-dimensional projected estimate. In this appendix, $N$ denotes the finite local batch size, corresponding to $m$ in the main text.

---

**Algorithm 2** OptFast-HGR Algorithm (Projected Approximation)

---

1: **Input:** Feature matrices $f, g \in \mathbb{R}^{N \times K}$, projection dimension $r \ll N$, regularization weight $\lambda$.
2: **Output:** OptFast-HGR objective $\mathcal{J}_{\text{OptFast}}$.
3: *// Step 1: Normalization*
4: Normalize $f, g$ as in Algorithm 1.
5: $\mathcal{J}_{\text{align}} = \frac{1}{N} \sum_{i=1}^{N} \cos(f_i, g_i)$
6: *// Step 2: Projected off-diagonal structure*
7: Draw a normalized random projection matrix $R \in \mathbb{R}^{N \times r}$.
8: Compute diagonal vectors $\mathbf{d}_f = \text{diag}(ff^\top)$ and $\mathbf{d}_g = \text{diag}(gg^\top)$.
9: Compute projected off-diagonal structures without storing the full Gram matrices:
10: $P_f = f(f^\top R) - \text{Diag}(\mathbf{d}_f)R, \quad P_g = g(g^\top R) - \text{Diag}(\mathbf{d}_g)R$
11: $\mathcal{J}_{\text{struct}} = \text{CosineSimilarity}(\text{flatten}(P_f), \text{flatten}(P_g))$
12: **return** $\mathcal{J}_{\text{OptFast}} = \mathcal{J}_{\text{align}} - \frac{\lambda}{2}\mathcal{J}_{\text{struct}}$

---

## B.1. Source of Approximation

Let $\widehat{\mathbf{G}}_f = \mathcal{D}_{\text{off}}(ff^\top), \widehat{\mathbf{G}}_g = \mathcal{D}_{\text{off}}(gg^\top)$ be the off-diagonal Gram structures after normalization. The full UniFast-HGR structural term compares the off-diagonal Gram structures, either row-wise or through their aggregate vectorized structure. OptFast-HGR draws a random projection matrix $\mathbf{R} \in \mathbb{R}^{N \times r}$ with $r \ll N$ and uses

$$
\begin{aligned}
\mathbf{P}_f &= \widehat{\mathbf{G}}_f \mathbf{R} = f(f^\top \mathbf{R}) - \text{Diag}(\mathbf{d}_f)\mathbf{R}, \\
\mathbf{P}_g &= \widehat{\mathbf{G}}_g \mathbf{R} = g(g^\top \mathbf{R}) - \text{Diag}(\mathbf{d}_g)\mathbf{R},
\end{aligned}
\tag{29}
$$

where $\mathbf{d}_f = \text{diag}(ff^\top)$ and $\mathbf{d}_g = \text{diag}(gg^\top)$. Under exact row normalization, $\mathbf{d}_f$ and $\mathbf{d}_g$ are close to all-one vectors, which gives the simplified implementation used in practice. This avoids storing the full Gram matrix during the structural comparison while preserving the same off-diagonal signal in the projected space.

## B.2. Projection Error

Assume that $\mathbf{R}$ is scaled so that $\mathbb{E}[\mathbf{R}\mathbf{R}^\top] = I_N$. For normalized random projections satisfying the usual inner-product preservation property, the projected Frobenius inner product concentrates around the full off-diagonal inner product:

$$
\left| \left\langle \widehat{\mathbf{G}}_f \mathbf{R}, \widehat{\mathbf{G}}_g \mathbf{R} \right\rangle_F - \left\langle \widehat{\mathbf{G}}_f, \widehat{\mathbf{G}}_g \right\rangle_F \right| \leq \epsilon_R \|\widehat{\mathbf{G}}_f\|_F \|\widehat{\mathbf{G}}_g\|_F.
\tag{30}
$$

The bound holds with high probability for a sufficiently large projection dimension $r$, with $\epsilon_R$ decreasing as $r$ increases. The same concentration controls the cosine-based structural score after normalization of the projected vectors:

$$
\mathcal{J}_{\text{struct}}^{\text{Opt}} = \text{CosineSimilarity}\left(\text{vec}(\mathbf{P}_f), \text{vec}(\mathbf{P}_g)\right).
\tag{31}
$$

## B.3. Computational Trade-Off

The full UniFast-HGR structural term forms and compares an $N \times N$ off-diagonal Gram structure. OptFast-HGR replaces this comparison with projected matrices of size $N \times r$. Since $f(f^\top \mathbf{R})$ can be computed through a $K \times r$ intermediate, the dominant structural computation is reduced to $O(NKr)$ when $r \ll N$, and the sketch memory is reduced to $O(Nr)$. The approximation preserves the same alignment term and TSS principle while reducing local-batch memory pressure.

## B.4. Empirical Validation

The trade-off is empirically favorable:

**Accuracy**: In ImageNet classification, the Top-1 accuracy difference between UniFast-HGR and OptFast-HGR is 0.5 percentage points.

**Stability**: The projected off-diagonal estimate keeps the structural loss bounded by cosine normalization and avoids the full Gram comparison when local batches are large.

## C. Asymptotic Complexity Comparison

This appendix compares the asymptotic complexity of UniFast-HGR and OptFast-HGR with earlier CCA/HGR-style methods and nonlinear dependence estimators. Table 7 also includes CKA, dCor, and recent $I_d$Cor as reference objectives.

UniFast-HGR uses non-iterative matrix operations to reduce covariance-space cost. Unlike Soft-HGR, it avoids direct construction of high-dimensional covariance matrices during the structural comparison. OptFast-HGR further reduces the off-diagonal structural comparison through random projection while maintaining a bounded cosine objective. Both versions remain fully differentiable and can be attached to multimodal neural backbones. Their main advantage is improved scaling with feature dimension $K$, which makes HGR-style dependence regularization practical for high-dimensional neural training.

*Table 7.* Overall complexity comparison analysis (local batch size $m$, feature dimension $K$, network layers $L$, projection dimension $r$).

| Methods | Time Complexity | Characteristic |
|---|---|---|
| CCA | $O(mK^2 + K^3)$ | Classic linear method with whitening |
| DeepCCA | $O(LmK^2 + K^3)$ | Nonlinear, covariance bottleneck |
| Soft CCA | $O(LmK^2)$ | Stochastic decorrelation objective |
| Soft-HGR | $O(mK^2)$ | Explicit $K \times K$ covariance interaction |
| CKA | $O(m^2K + m^2)$ | Gram-based structural similarity |
| dCor | $O(m^2K)$ | Nonlinear dependence via distances |
| $I_d$Cor | $O(m^2K)$ | Intrinsic-dimension-aware comparison |
| UniFast-HGR | $O(m^2K)$ | Full off-diagonal Gram structure |
| OptFast-HGR | $O(mKr)$ | Projected off-diagonal structure |

*Table 8.* Comparison of various methods on the Berlin HSI-SAR dataset (%).

| Class | CCA | Deep CCA | Soft CCA | Dot Product | Cosine Similarity | Soft-HGR | UniFast-HGR | OptFast-HGR |
|---|---|---|---|---|---|---|---|---|
| OA | 70.93 | 72.74 | 71.54 | 75.20 | 75.51 | 65.80 | **80.75** | 80.46 |
| AA | 64.35 | 65.08 | 61.14 | 66.22 | 65.53 | 64.30 | **71.53** | 71.51 |
| Kappa | 58.28 | 60.23 | 58.33 | 62.77 | 62.53 | 52.99 | **70.44** | 70.21 |
| Forest | 81.90 | 64.17 | 87.16 | 76.68 | 79.92 | 67.54 | **87.61** | 82.18 |
| Residential area | 72.81 | 76.38 | 75.59 | 82.57 | 85.63 | 63.87 | **86.85** | 85.10 |
| Industrial area | 23.05 | **76.00** | 53.61 | 48.15 | 49.11 | 64.07 | 40.20 | 62.67 |
| Low plants | 71.44 | 89.08 | 62.68 | 65.08 | 54.31 | 82.05 | 73.70 | **89.23** |
| Soil | 85.97 | 72.10 | 78.01 | 82.53 | 82.88 | **88.16** | 82.42 | 78.63 |
| Allotment | 69.87 | 58.73 | 51.72 | **70.73** | 69.07 | 55.79 | 65.35 | 65.65 |
| Commercial area | **56.76** | 20.40 | 42.81 | 35.88 | 23.77 | 37.97 | 54.30 | 27.61 |
| Water | 52.98 | 63.78 | 37.53 | 68.15 | 79.58 | 54.95 | **81.85** | 81.01 |

## D. Detailed Experimental Results

### D.1. Image Classification

Table 8 presents the detailed comparative results of remote sensing data classification with a dual-channel deep learning framework and a ResNet-50 backbone on the Berlin dataset. Table 9 reports the corresponding results with a dual-channel visual transformer framework on the Houston 2018 dataset.

The results show that UniFast-HGR and OptFast-HGR improve the overall classification performance over traditional CCA and similarity-based objectives. This indicates that the proposed objectives capture cross-modal relations effectively on both the Berlin HSI-SAR and Houston 2018 HSI-LiDAR datasets, under both CNN and transformer frameworks.

Traditional CCA is less effective at capturing nonlinear relations in remote sensing data. Deep CCA improves over CCA in most aggregate metrics, showing the benefit of nonlinear feature mappings. Dot product and cosine similarity remain strong lightweight baselines on these datasets. Soft-HGR underperforms in several OA metrics, which is consistent with the instability of covariance-based objectives under high-dimensional or limited-batch settings.

To evaluate computational efficiency, the experiments compare the execution time of remote sensing data classification on the Berlin and Houston 2018 datasets, using a dual-channel deep learning framework with ResNet-50 and a dual-channel visual transformer framework, respectively. The results in Table 10 show that CCA, Deep CCA, and Soft CCA require longer execution time. UniFast-HGR and OptFast-HGR keep the runtime close to dot product and cosine similarity while providing higher accuracy.

*Table 9.* Comparison of various methods on the Houston 2018 HSI-LiDAR dataset (%).

| Class | CCA | Deep CCA | Soft CCA | Dot Product | Cosine Similarity | Soft HGR | UniFast HGR | OptFast HGR |
|---|---|---|---|---|---|---|---|---|
| OA | 88.28 | 89.82 | 88.81 | 91.59 | 92.04 | 85.86 | **93.65** | 93.25 |
| AA | 92.20 | 93.92 | 93.14 | 93.85 | 94.67 | 91.01 | **96.15** | 95.71 |
| Kappa | 84.89 | 86.89 | 85.62 | 89.13 | 89.65 | 81.91 | **91.77** | 91.25 |
| Healthy grass | 95.62 | 97.84 | 97.97 | 78.15 | 98.24 | **98.76** | 95.18 | 97.66 |
| Stressed grass | 86.77 | 83.27 | 89.16 | **97.58** | 89.66 | 83.84 | 93.57 | 93.27 |
| Artificial turf | **100.00** | 99.83 | **100.00** | **100.00** | **100.00** | **100.00** | **100.00** | **100.00** |
| Evergreen trees | 99.05 | 98.28 | 97.81 | 96.15 | 98.95 | 97.80 | **99.37** | 98.45 |
| Deciduous trees | 96.05 | 95.18 | 95.92 | 94.94 | 97.57 | 96.69 | **98.75** | 98.01 |
| Bare earth | **100.00** | **100.00** | **100.00** | 99.99 | **100.00** | 99.99 | **100.00** | 99.99 |
| Water | **100.00** | **100.00** | **100.00** | **100.00** | **100.00** | **100.00** | **100.00** | **100.00** |
| Residential buildings | 94.02 | 97.90 | 97.42 | 96.88 | 91.92 | **98.49** | 97.04 | 98.20 |
| Non-residential buildings | 94.80 | 94.53 | 93.48 | 95.92 | 97.47 | 91.40 | **98.89** | 96.86 |
| Road | 56.85 | 69.52 | 62.37 | 74.35 | 69.20 | 50.99 | **82.82** | 79.26 |
| Sidewalks | 81.24 | 78.02 | 71.27 | 73.72 | **83.17** | 65.75 | 82.75 | 78.53 |
| Crosswalks | 76.18 | 95.93 | 87.92 | 91.78 | 91.40 | 74.92 | **96.82** | 92.96 |
| Major thoroughfares | 73.24 | 79.62 | 82.78 | 85.45 | 86.32 | 78.80 | 85.47 | **87.16** |
| Highways | 98.90 | 95.04 | 96.08 | 97.65 | 99.47 | 96.73 | 98.24 | **99.67** |
| Railways | 99.77 | 99.87 | 99.87 | 99.60 | 99.50 | 99.40 | **99.94** | 99.90 |
| Paved parking lots | 92.95 | 96.88 | 94.18 | **97.46** | 92.83 | 93.98 | 97.02 | 95.53 |
| Unpaved parking lots | **100.00** | **100.00** | **100.00** | **100.00** | **100.00** | 94.07 | **100.00** | **100.00** |
| Cars | 99.13 | 97.41 | 97.17 | 97.45 | 97.65 | 98.53 | **99.16** | 98.70 |
| Trains | 99.95 | 99.41 | 99.57 | **100.00** | **100.00** | **100.00** | 99.99 | **100.00** |
| Stadium seats | 99.57 | 99.94 | 99.83 | **100.00** | **100.00** | **100.00** | 99.98 | **100.00** |

## D.2. Remote Sensing Semantic Segmentation

The detailed semantic segmentation results on Vaihingen and Globe230k are shown in Table 11 and Table 12, respectively. Figure 3 shows qualitative segmentation examples on the Vaihingen dataset. UniFast-HGR and OptFast-HGR better preserve long-range semantic regions and object boundaries, which leads to more accurate semantic segmentation of remote sensing imagery.

## D.3. Multimodal Emotion Recognition

The emotion recognition experiments on the IEMOCAP dataset are reported in Table 13. UniFast-HGR and OptFast-HGR improve both W-F1 and ACC over conventional CCA and similarity-based objectives. These results indicate that the proposed objectives capture cross-modal emotion cues more reliably than the compared correlation and similarity objectives.

## D.4. Image Classification on CIFAR-100

To evaluate the scalability of UniFast-HGR and OptFast-HGR on standard visual classification tasks, experiments were conducted on the CIFAR-100 dataset using five representative backbone architectures: ViT-B/32, ResNet50, CLIP, SigLIP, and DINOv2. The comparison includes nonlinear correlation baselines (CKA, dCor, $I_d$Cor) and Soft-HGR, reporting top-1 accuracy (%) in Table 14.

*Table 10.* Execution time comparison on the Berlin and Houston 2018 datasets (s/epoch).

| Method | ResNet-50 | | Vision Transformer | |
|---|---|---|---|---|
| | Berlin | Houston 2018 | Berlin | Houston 2018 |
| CCA | 2967.52 | – | 307.82 | 1243.23 |
| Deep CCA | 250.51 | 1158.42 | 379.82 | 1520.09 |
| Soft CCA | 314.93 | 1751.98 | 211.03 | 929.50 |
| Dot Product | 23.18 | 106.05 | 20.85 | 48.89 |
| Cosine Similarity | 23.40 | 106.14 | 20.93 | 49.34 |
| Soft-HGR | 25.83 | 110.53 | 21.62 | 58.03 |
| UniFast-HGR | 24.53 | 108.56 | 21.23 | 57.00 |
| OptFast-HGR | 23.54 | 106.27 | 21.02 | 52.41 |

*Table 11.* Comparison of various methods on the Vaihingen dataset (%).

| Class | CCA | Deep CCA | Soft CCA | Dot Product | Cosine Similarity | Soft HGR | UniFast HGR | OptFast HGR |
|---|---|---|---|---|---|---|---|---|
| OA | 91.15 | 91.39 | 91.41 | 92.61 | 92.56 | 90.10 | **93.01** | 92.95 |
| mIoU | 79.37 | 81.35 | 81.44 | 83.65 | 83.34 | 76.87 | **84.62** | 84.57 |
| Imp. | 91.43 | 92.57 | 92.52 | **94.97** | 93.38 | 91.39 | 93.62 | 93.47 |
| Building | 97.37 | 96.94 | 97.19 | 95.55 | 97.62 | 95.93 | 97.86 | **97.92** |
| Low. | 80.19 | 79.51 | 79.62 | 80.36 | 81.94 | 73.08 | **82.03** | 81.86 |
| Tree | 91.03 | 91.53 | 91.24 | **94.93** | 92.67 | 93.41 | 93.82 | 93.79 |
| Car | 76.94 | 82.94 | 83.76 | 83.41 | 88.53 | 73.86 | **90.15** | 89.95 |

*Table 12.* Comparison of various methods on the Globe230k dataset (%).

| Class | CCA | Deep CCA | Soft CCA | Dot Product | Cosine Similarity | Soft HGR | UniFast HGR | OptFast HGR |
|---|---|---|---|---|---|---|---|---|
| OA | 87.92 | 88.27 | 87.60 | 90.92 | 90.81 | 86.46 | **91.48** | 91.23 |
| mIoU | 67.49 | 67.85 | 66.71 | 75.67 | 75.53 | 64.82 | **76.36** | 76.15 |
| Cropland | 83.27 | 91.86 | 79.12 | 89.76 | 90.19 | 91.75 | **92.15** | 90.32 |
| Forest | 91.60 | 95.51 | 90.20 | 95.24 | 96.32 | 93.46 | 96.73 | **96.89** |
| Grassland | 58.75 | 65.44 | 61.48 | 79.93 | 78.47 | 54.83 | **80.68** | 80.31 |
| Shrubland | 62.49 | 73.07 | 55.34 | 72.89 | 71.50 | 57.63 | **75.41** | 72.62 |
| Wetland | 73.08 | 71.80 | 42.76 | 77.54 | 76.72 | 42.09 | 77.92 | **78.49** |
| Water | 85.22 | 89.62 | 90.83 | 94.65 | 94.26 | 83.69 | **95.62** | 95.35 |
| Tundra | 9.31 | 0.00 | 5.32 | 38.58 | 36.82 | 0.00 | **43.07** | 41.27 |
| Impervious surface | 80.92 | 86.59 | 81.50 | 93.17 | 92.90 | 80.78 | 93.50 | **94.10** |
| Bareland | 72.43 | 87.37 | 74.57 | 91.10 | 90.64 | 73.15 | **91.46** | 91.07 |
| Ice/ snow | 91.25 | 97.53 | 91.82 | 97.62 | 98.21 | 90.76 | **98.39** | 97.85 |

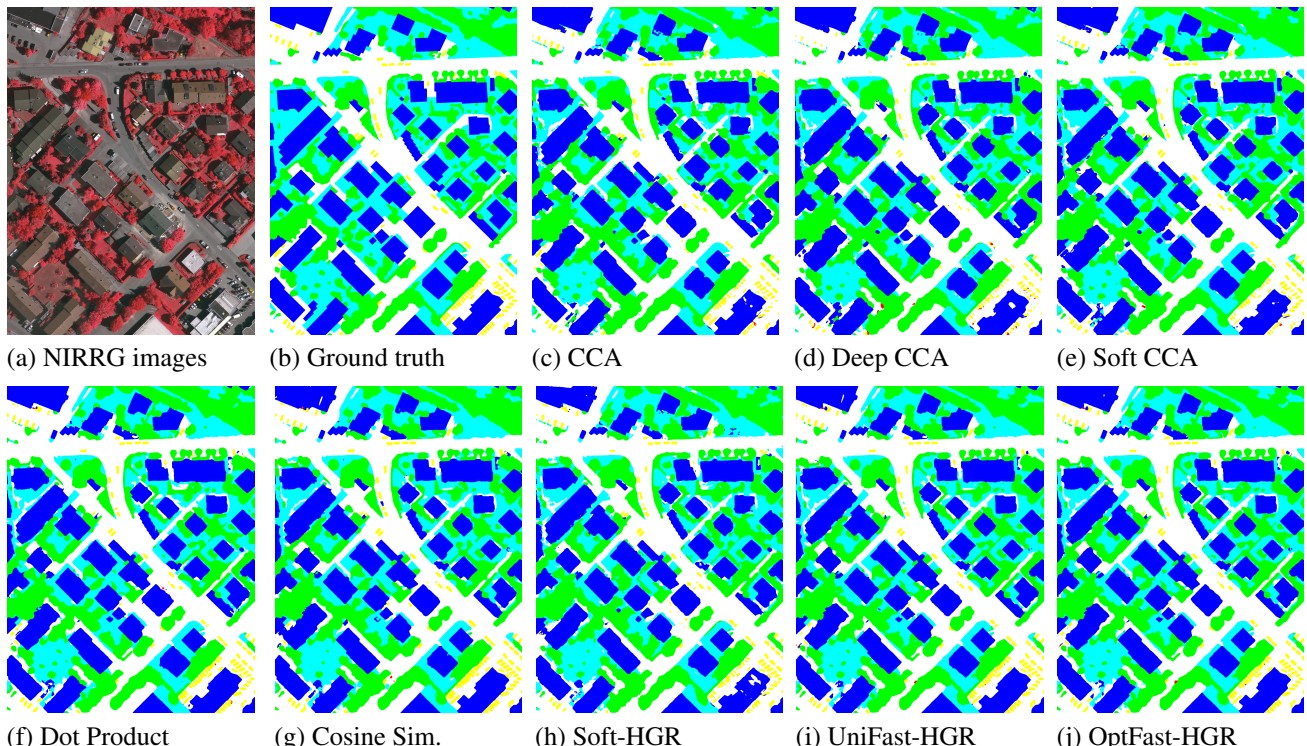

(a) NIRRG images     (b) Ground truth     (c) CCA     (d) Deep CCA     (e) Soft CCA

(f) Dot Product     (g) Cosine Sim.     (h) Soft-HGR     (i) UniFast-HGR     (j) OptFast-HGR

*Figure 3.* Qualitative segmentation results on the Vaihingen test set.

*Table 13.* Comparison of various methods on the IEMOCAP dataset (%).

| Class | CCA | Deep CCA | Soft CCA | Dot Product | Cosine Similarity | Soft-HGR | UniFast-HGR | OptFast-HGR |
|---|---|---|---|---|---|---|---|---|
| W-F1 | 67.51 | 67.82 | 68.57 | 69.87 | 69.60 | 71.43 | **73.57** | 73.32 |
| ACC | 67.41 | 67.78 | 68.58 | 70.14 | 69.50 | 71.29 | **73.66** | 73.43 |
| Happy | 50.77 | 49.81 | 46.77 | 50.51 | 53.85 | 54.92 | **66.63** | 59.67 |
| Sad | 79.65 | 81.82 | 79.29 | 81.96 | 81.39 | 81.53 | 84.79 | **85.23** |
| Neutral | 68.11 | 69.58 | 69.59 | 71.24 | 71.89 | 70.84 | **74.30** | 73.00 |
| Angry | 61.98 | 62.53 | 64.60 | 65.90 | 65.82 | 70.32 | 70.46 | **71.04** |
| Excited | 76.70 | 76.56 | 75.00 | 74.48 | 74.91 | 75.00 | **77.14** | 77.09 |
| Frustrated | 60.66 | 59.35 | 65.62 | 67.32 | 63.17 | 69.45 | **71.22** | 70.36 |

UniFast-HGR consistently outperforms all baselines across architectures. For ResNet50, it achieves 76.8% accuracy, 2.3 points higher than the baseline (74.5%) and 1.0 point higher than the strongest baseline ($I_d$Cor, 75.8%). On ViT-B/32, UniFast-HGR reaches 86.4%, surpassing $I_d$Cor (86.1%) and Soft-HGR (85.5%) by 0.3 and 0.9 points, respectively. With state-of-the-art vision models, UniFast-HGR achieves the best performance: 88.9% on SigLIP and 89.3% on DINOv2.

OptFast-HGR balances efficiency and accuracy, with results close to UniFast-HGR. For instance, on DINOv2, OptFast-HGR reaches 88.5%, which is 0.8 points below UniFast-HGR and remains competitive with the strongest baselines. Across all models, the accuracy gap between UniFast-HGR and OptFast-HGR remains within 1.0 point. These results support the use of OptFast-HGR as a low-memory approximation with limited accuracy loss.

### D.5. Comparison with State-of-the-Art Foundation Models

The comparison evaluates whether HGR-style dependence regularization improves feature alignment in large visual and multimodal backbones. The objective is integrated into CLIP (ViT-B/32), SigLIP, and DINOv2 (ViT-L/14), and ImageNet embedding similarities are measured across models. Table 15 reports the resulting correlation scores together with dCor, $I_d$Cor, and Soft-HGR. UniFast-HGR improves the correlation score when attached to the base model, supporting its role as an auxiliary alignment objective.

*Table 14.* Experimental results on CIFAR-100.

| Dataset | Model | Baseline | CKA | dCor | $I_d$Cor | Soft-HGR | UniFast-HGR | OptFast-HGR |
|---------|-------|----------|-----|------|----------|----------|-------------|-------------|
| | ViT-B/32 | 85.3 | 85.6 | 85.8 | 86.1 | 85.5 | **86.4** | 86.2 |
| CIFAR-100 | ResNet50 | 74.5 | 75.1 | 75.2 | 75.8 | 75.3 | **76.8** | 76.1 |
| Accuracy (%) | CLIP | 80.5 | 81.2 | 81.4 | 81.6 | 81.3 | **82.5** | 81.5 |
| | SigLIP | 87.1 | 87.5 | 87.8 | 88.2 | 87.4 | **88.9** | 88.4 |
| | DINOv2 | 87.5 | 87.8 | 88.3 | 88.7 | 87.7 | **89.3** | 88.5 |

# E. Ablation Studies

Table 16 reports ablations across remote sensing classification, semantic segmentation, and multimodal emotion recognition benchmarks. The results complement the component-wise progression in Table 6 by showing that the same design choices remain effective beyond Berlin and ImageNet-1K.

Variance constraints are important for numerical stability. Without them, the objective becomes more sensitive to feature scale and produces a clear performance drop. On Berlin, removing variance constraints reduces OA/AA to 68.53%/67.26%, compared with 80.75%/71.53% for UniFast-HGR. Similar degradation appears on Houston 2018, Vaihingen, Globe230k, and IEMOCAP. This supports the role of variance control in preventing magnitude-driven solutions and stabilizing feature learning.

Diagonal suppression also contributes consistently. In normalized Gram structures, diagonal entries correspond to fixed self-similarities and do not encode relations between distinct samples. Retaining these entries weakens the structural signal. For example, the "w/ Main Diagonal" variant decreases Berlin OA from 80.75% to 80.62% and Houston 2018 OA from 93.65% to 93.46%. The gain is moderate at large batch sizes, but the same pattern is consistent across datasets. This agrees with the sensitivity results in Section 4.7, where diagonal removal is more useful when off-diagonal diversity is limited.

The covariance-based variant remains competitive but is consistently weaker than the final Gram-space formulation. On Berlin, it reaches 79.83% OA, compared with 80.75% for UniFast-HGR. The same trend holds for Houston 2018, Vaihingen, Globe230k, and IEMOCAP. These results indicate that the cosine/Gram formulation preserves useful dependence information while avoiding the numerical and computational burden of explicit covariance operations.

OptFast-HGR gives a stable speed–accuracy trade-off. It remains close to the full objective across all datasets. On ImageNet-1K, it reaches 79.6% Top-1 compared with 80.1% for UniFast-HGR in Table 6. In Table 16, it also stays within a small margin of the full method on Berlin, Houston 2018, Vaihingen, Globe230k, and IEMOCAP. This behavior is consistent with its role as a low-memory approximation of the off-diagonal structural term.

The ablation results show that each design choice contributes to the final behavior. Variance constraints stabilize optimization, diagonal suppression reduces self-correlation bias, the cosine/Gram formulation improves computational practicality, and OptFast-HGR provides a low-memory approximation with limited accuracy loss. The full UniFast-HGR objective gives the best overall balance across the evaluated datasets and metrics.

# F. Sensitivity and MI-Estimator Comparisons

### F.1. Batch-Size Sensitivity

Table 17 gives an ImageNet-1K sweep over local batch size. Smaller batches increase the relative diagonal mass and sampling noise in Gram-based estimators. TSS reduces this effect by removing invariant self-correlation before computing the structural term.

### F.2. Batch-Composition Sensitivity

Table 18 evaluates skewed, homogeneous, and near-duplicate batches on Berlin. Extreme homogeneity reduces useful off-diagonal diversity for any batch-structure estimator. TSS does not remove this boundary condition, but it keeps the degradation smaller than Soft-HGR by suppressing the trivial diagonal mode.

*Table 15.* Correlation results on ImageNet representations.

| Methods | Models | EfficientNet | ResNet50 | ViT-B/32 | CLIP | SigLIP | DINOv2 |
|---|---|---|---|---|---|---|---|
| dCor | EfficientNet | 1. | 0.45 | 0.42 | 0.29 | 0.34 | 0.41 |
| | ResNet50 | 0.45 | 1. | 0.43 | 0.54 | 0.58 | 0.56 |
| | ViT-B/32 | 0.42 | 0.43 | 1. | 0.46 | 0.49 | 0.48 |
| | CLIP | 0.29 | 0.54 | 0.46 | 1. | 0.82 | 0.78 |
| | SigLIP | 0.34 | 0.58 | 0.49 | 0.82 | 1. | 0.80 |
| | DINOv2 | 0.41 | 0.56 | 0.48 | 0.78 | 0.80 | 1. |
| $I_d$Cor | EfficientNet | 1. | 0.91 | 0.85 | 0.77 | 0.81 | 0.82 |
| | ResNet50 | 0.91 | 1. | 0.86 | 0.80 | 0.83 | 0.81 |
| | ViT-B/32 | 0.85 | 0.86 | 1. | 0.92 | 0.92 | 0.90 |
| | CLIP | 0.77 | 0.80 | 0.92 | 1. | 0.91 | 0.89 |
| | SigLIP | 0.81 | 0.83 | 0.92 | 0.91 | 1. | 0.92 |
| | DINOv2 | 0.82 | 0.81 | 0.90 | 0.89 | 0.92 | 1. |
| Soft-HGR | EfficientNet | 1. | 0.63 | 0.61 | 0.55 | 0.57 | 0.60 |
| | ResNet50 | 0.63 | 1. | 0.62 | 0.71 | 0.74 | 0.73 |
| | ViT-B/32 | 0.61 | 0.62 | 1. | 0.66 | 0.70 | 0.68 |
| | CLIP | 0.55 | 0.71 | 0.66 | 1. | 0.85 | 0.82 |
| | SigLIP | 0.57 | 0.74 | 0.70 | 0.85 | 1. | 0.85 |
| | DINOv2 | 0.60 | 0.73 | 0.68 | 0.82 | 0.85 | 1. |
| UniFast-HGR | EfficientNet | 1. | 0.92 | 0.87 | 0.84 | 0.87 | 0.86 |
| | ResNet50 | 0.92 | 1. | 0.86 | 0.85 | 0.88 | 0.84 |
| | ViT-B/32 | 0.87 | 0.86 | 1. | 0.93 | 0.94 | 0.92 |
| | CLIP | 0.84 | 0.85 | 0.93 | 1. | 0.92 | 0.91 |
| | SigLIP | 0.87 | 0.88 | 0.94 | 0.92 | 1. | 0.94 |
| | DINOv2 | 0.86 | 0.84 | 0.92 | 0.91 | 0.94 | 1. |
| OptFast-HGR | EfficientNet | 1. | 0.91 | 0.84 | 0.82 | 0.83 | 0.83 |
| | ResNet50 | 0.91 | 1. | 0.85 | 0.82 | 0.83 | 0.83 |
| | ViT-B/32 | 0.84 | 0.85 | 1. | 0.91 | 0.92 | 0.91 |
| | CLIP | 0.82 | 0.82 | 0.91 | 1. | 0.91 | 0.90 |
| | SigLIP | 0.83 | 0.83 | 0.92 | 0.91 | 1. | 0.92 |
| | DINOv2 | 0.83 | 0.83 | 0.91 | 0.90 | 0.92 | 1. |

## F.3. Regularization Weight

Table 19 shows stable behavior over $\lambda \in [0.1, 1.0]$. In practice, $\lambda = 0.5$ is a reliable default for most tasks, while $\lambda = 1.0$ is useful when the supervised loss dominates the early training stage.

## F.4. Low-Dimensional and Pair-Weighting Analyses

The fixed variance constraint is designed for high-dimensional features, where covariance whitening is unstable. In very small feature spaces, strict normalization can suppress useful task-specific variation. Table 20 shows that an adaptive constraint tied to intrinsic dimensionality improves the smallest $K$ settings and matches the fixed version once $K$ becomes sufficiently large.

A relevance-weighted extension is also compatible with the pairwise aggregation objective. It gives small gains on MSR-VTT (R@1: 43.3 to 43.8) and IEMOCAP (W-F1/ACC: 73.57/73.66 to 73.89/74.02). The default protocol uses uniform aggregation to keep the objective parameter-free and avoid task-specific tuning.

## F.5. Comparison with Neural MI Estimators

Table 21 compares UniFast-HGR with standard neural mutual-information estimators (Belghazi et al., 2018; Nguyen et al., 2010; Cheng et al., 2020) under matched backbones and training budgets. UniFast-HGR gives higher accuracy without adding a separate critic network.

*Table 16.* Ablation study across multiple datasets (%).

| Methods | Berlin | | Houston 2018 | | Vaihingen | | Globe230k | | IEMOCAP | |
|---|---|---|---|---|---|---|---|---|---|---|
| | OA | AA | OA | AA | OA | mIoU | OA | mIoU | W-F1 | ACC |
| w/o Variance Constraints | 68.53 | 67.26 | 86.72 | 92.24 | 90.82 | 77.55 | 87.41 | 66.96 | 71.62 | 71.49 |
| w/ Main Diagonal | 80.62 | 71.39 | 93.46 | 95.97 | 92.85 | 84.57 | 91.32 | 76.27 | 73.41 | 73.38 |
| Covariance-based | 79.83 | 70.92 | 92.87 | 95.43 | 92.26 | 83.89 | 90.75 | 75.64 | 72.95 | 72.87 |
| OptFast-HGR | 80.46 | 71.51 | 93.25 | 95.71 | 92.95 | 84.57 | 91.23 | 76.15 | 73.32 | 73.43 |
| UniFast-HGR | **80.75** | **71.53** | **93.65** | **96.15** | **93.01** | **84.62** | **91.48** | **76.36** | **73.57** | **73.66** |

*Table 17.* Batch-size sensitivity on ImageNet-1K Top-1 accuracy (%).

| Local batch size $m$ | 16 | 32 | 64 | 128 | 256 |
|---|---|---|---|---|---|
| Soft-HGR | 72.1 | 73.5 | 74.8 | 75.7 | 76.3 |
| UniFast-HGR, diagonal retained | 75.3 | 76.7 | 77.9 | 78.6 | 79.8 |
| UniFast-HGR with TSS | **78.2** | **79.0** | **79.5** | **79.8** | **80.1** |

## G. Computational Efficiency

To evaluate the impact of feature dimension and local batch size on computational performance, the execution times of UniFast-HGR, OptFast-HGR, and baseline methods (CCA, Deep CCA, Soft CCA, CKA, dCor, $I_d$Cor, and Soft-HGR) were measured using randomly generated tensor pairs $(f, g) \in \mathbb{R}^{m \times K}$. Each method's correlation computation was repeated 10,000 times across local batch sizes $m \in \{16, 32, 64, 128, 256\}$ and feature dimensions $K \in \{10, 50, 100, 150, 200, 300, 400, 500\}$. The average execution times are visualized in Figure 4.

Figure 4 presents runtime comparisons across four representative local batch sizes ($m = 16, 64, 128, 256$):

1. **OptFast-HGR: Low Runtime Growth with Feature Dimension.** At $m = 256$, OptFast-HGR shows a gentle increase in runtime from 0.000265 seconds ($K = 10$) to 0.000877 seconds ($K = 500$), indicating near-linear scaling in the tested range due to vectorized projected operations. This represents a clear speedup over CCA, which has substantially higher runtime because of covariance estimation and matrix decomposition.

2. **UniFast-HGR: Balanced Efficiency and Accuracy.** UniFast-HGR has slightly higher runtime than OptFast-HGR due to its full Gram-space structural computation, but it still remains faster than covariance-based and kernel-based baselines in most settings. At $m = 128$, its runtime increases from 0.000419 seconds ($K = 10$) to 0.000537 seconds ($K = 500$), while CCA grows more sharply. This behavior is consistent with the method's dependence on local-batch Gram operations rather than high-dimensional covariance whitening.

3. **Baseline Methods: Higher Cost under Scaling.** CCA grows rapidly with feature dimension due to covariance decomposition. dCor grows mainly with pairwise sample distances and becomes slower as local batch size increases. Soft CCA and CKA also incur higher costs due to covariance or kernel computations. $I_d$Cor is the slowest in this benchmark, with runtime exceeding 0.2 seconds at $m = 256$, $K = 500$. These trends show that the computational bottleneck is not only the feature dimension, but also the way each method forms and normalizes pairwise or covariance statistics.

4. **Local-Batch Robustness.** At smaller local batch sizes ($m = 16$), OptFast-HGR maintains the fastest execution time, while CKA and Soft-HGR show modest batch-size sensitivity but higher absolute runtime. As local batch size increases, the performance gap between UniFast/OptFast-HGR and the baseline methods becomes more visible. The projected estimate in OptFast-HGR is especially useful when the local feature block becomes large.

Figure 5 further compares execution times for a fixed feature dimension ($K = 300$) across varying local batch sizes. OptFast-HGR consistently achieves low execution time with only a slight increase as $m$ grows. UniFast-HGR also shows relatively low execution time, increasing more slowly than many baseline methods. CCA rises significantly with increasing local batch size, reflecting the cost of covariance operations. dCor and $I_d$Cor also increase rapidly with local batch size. In contrast, OptFast-HGR and UniFast-HGR show flatter execution-time curves, indicating stable efficiency across different local batch sizes.

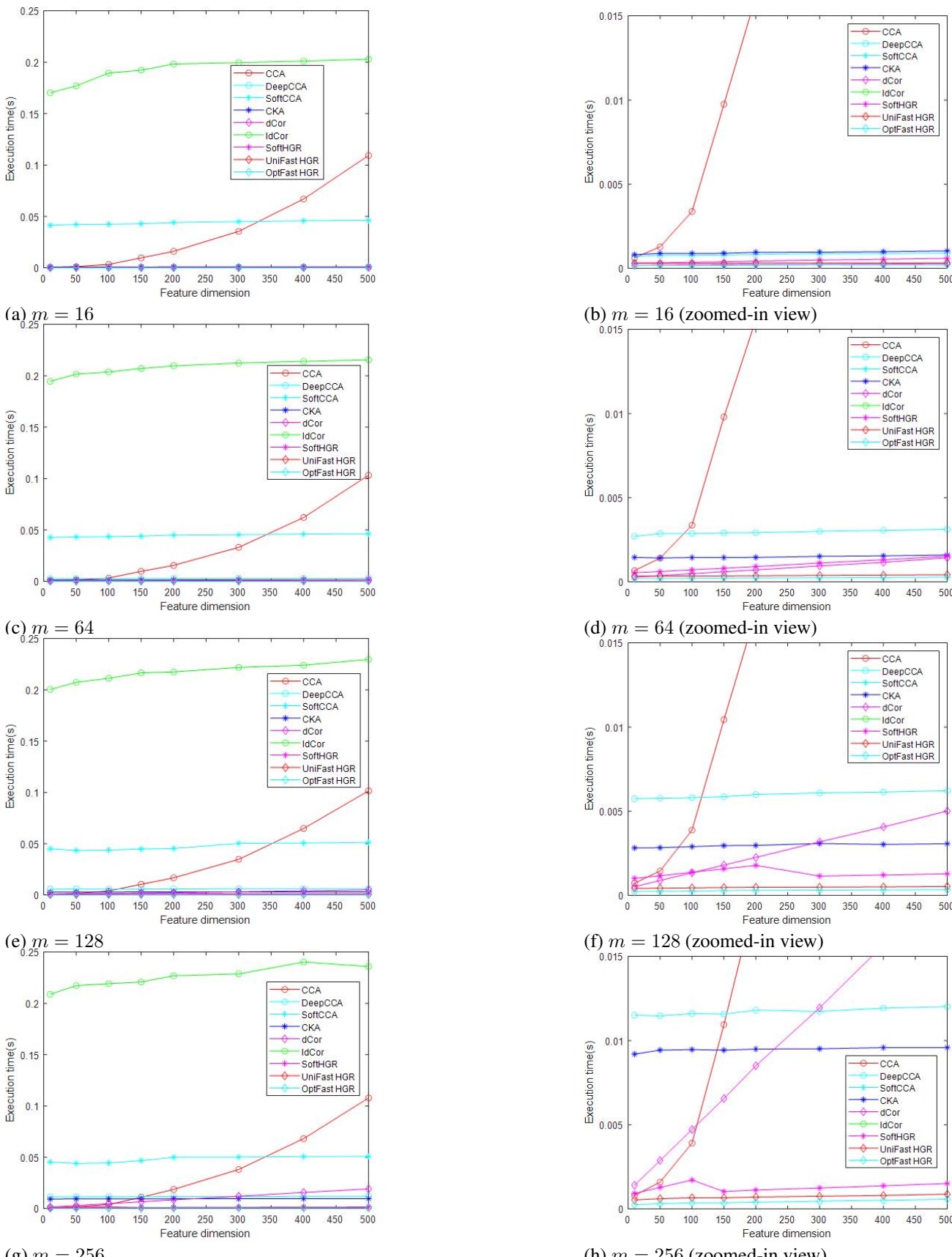

(a) $m = 16$

(b) $m = 16$ (zoomed-in view)

(c) $m = 64$

(d) $m = 64$ (zoomed-in view)

(e) $m = 128$

(f) $m = 128$ (zoomed-in view)

(g) $m = 256$

(h) $m = 256$ (zoomed-in view)

*Figure 4.* Execution time comparison for correlation methods across local batch sizes and feature dimensions.

*Table 18.* Batch-composition sensitivity on Berlin OA (%).

| Sampler | Soft-HGR | UniFast-HGR |
|---|---|---|
| Standard random | 65.80 | **80.75** |
| 80% single-class imbalance | 59.20 | **78.90** |
| Single-label homogeneous | 52.40 | **77.20** |
| 30% near-duplicate injected | 63.10 | **79.82** |

*Table 19.* Sensitivity to the regularization weight $\lambda$.

| Dataset | Best $\lambda$ | Range for $\lambda \in [0.1, 1.0]$ | Main metric |
|---|---|---|---|
| Berlin | 0.5 | 80.50–80.75 | OA |
| ImageNet-1K | 0.5 | 79.80–80.10 | Top-1 |
| COCO | 0.5 | 41.90–42.10 | R@1 |
| IEMOCAP | 1.0 | 73.40–73.66 | ACC |
| Vaihingen | 0.5 | 92.90–93.01 | OA |

The efficiency of UniFast-HGR and OptFast-HGR mainly comes from avoiding high-dimensional covariance decomposition and from using normalized Gram-space operations. UniFast-HGR evaluates the full off-diagonal structure, while OptFast-HGR replaces it with a projected sketch. This removes the main whitening or decomposition step and reduces the dominant computational cost. These results show that UniFast-HGR and OptFast-HGR improve runtime over traditional correlation methods while preserving accuracy, which is useful for multimodal tasks with tight computational budgets.

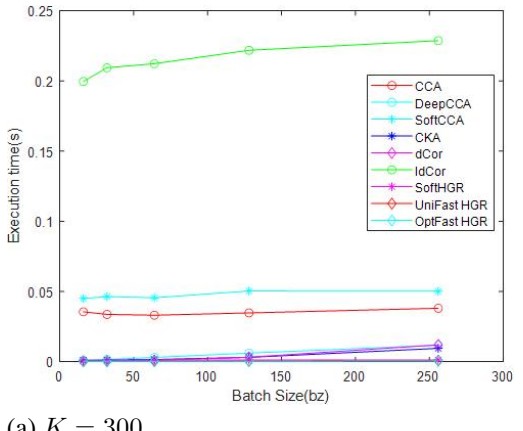

(a) $K = 300$

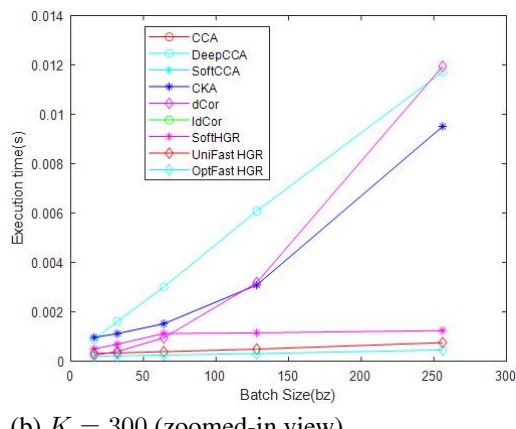

(b) $K = 300$ (zoomed-in view)

*Figure 5.* Execution time comparison across methods for fixed feature dimension $K = 300$ with varying local batch sizes.

## H. Robustness to Real-World Challenges

To evaluate the robustness of UniFast-HGR in practical stress settings, experiments were conducted under three challenging conditions: high noise, modality imbalance, and spurious correlations. Protocols for each dataset use IEMOCAP for audio noise, Flickr30K for modality imbalance, and Berlin for spurious correlations. The evaluation follows the same task metrics as the corresponding benchmarks.

**High Noise Perturbation:** Gaussian noise with a standard deviation of 30% was added to audio features in the IEMOCAP dataset. Performance was compared against CLIP-based fusion and Soft-HGR. CLIP-based fusion achieved 65.8% accuracy, a 5.5-point decrease from clean data. Soft-HGR dropped to 63.1% accuracy, an 8.2-point reduction. UniFast-HGR retained 70.2% accuracy, with a 3.5-point decline, outperforming CLIP-based fusion and Soft-HGR by 4.4 and 7.1 points, respectively.

**Modality Imbalance:** To simulate extreme text-pair scarcity, only 10% of paired text supervision was retained in a highly imbalanced Flickr30K training setting. CLIP and Deep CCA were evaluated as baselines. CLIP achieved 62.3% Recall@1,

*Table 20.* Low-dimensional boundary analysis with adaptive variance constraints on CIFAR-10 Top-1 accuracy (%).

| Feature dimension $K$ | 10 | 32 | 64 | 128 |
|---|---|---|---|---|
| Fixed constraint | 82.3 | 88.7 | 92.1 | 93.5 |
| Adaptive constraint | **85.1** | **90.2** | **92.2** | 93.5 |

*Table 21.* Matched comparison with neural MI estimators.

| Task | Backbone | InfoNCE | MINE | NWJ | CLUB | UniFast-HGR |
|---|---|---|---|---|---|---|
| ImageNet Top-1 (%) | ViT-B/32 | 76.6 | 77.2 | 76.9 | 77.5 | **80.1** |
| COCO R@1 (%) | CLIP ViT-B/32 | 38.9 | 39.5 | 39.2 | 39.8 | **42.1** |
| IEMOCAP ACC (%) | MultiEMO | 70.14 | 70.8 | 70.5 | 71.2 | **73.66** |

limited by reliance on paired text-image supervision. Deep CCA reached 59.7% Recall@1. UniFast-HGR attained 68.9% Recall@1, exceeding CLIP and Deep CCA by 6.6 and 9.2 points, respectively.

**Spurious Correlations:** To test resistance to misleading associations, 20% of training labels in the Berlin dataset were corrupted to introduce false building-to-forest mappings. Soft-HGR achieved 69.2% overall accuracy (OA), indicating sensitivity to spurious pairs under covariance-based dependence estimation. UniFast-HGR reached 77.3% OA, giving an 8.1-point improvement over Soft-HGR.

*Table 22.* Performance under real-world challenge settings.

| Scenario | Method | Metric | Value |
|---|---|---|---|
| High Noise (IEMOCAP) | CLIP-based fusion | Accuracy (%) | 65.8 |
| | Soft-HGR | Accuracy (%) | 63.1 |
| | UniFast-HGR | Accuracy (%) | 70.2 |
| Modality Imbalance (Flickr30K) | CLIP | Recall@1 (%) | 62.3 |
| | Deep CCA | Recall@1 (%) | 59.7 |
| | UniFast-HGR | Recall@1 (%) | 68.9 |
| Spurious Correlations (Berlin) | Soft-HGR | OA (%) | 69.2 |
| | UniFast-HGR | OA (%) | 77.3 |

These results show that UniFast-HGR maintains stronger performance under noise, text-pair scarcity, and spurious correlations. Its variance control, diagonal suppression, and normalized structural comparison help prioritize stable cross-modal relations under these stress conditions. The results also indicate that the objective does not rely only on clean paired supervision, but remains useful when part of the observed dependence is noisy or misleading.

## I. Discussion

The results support the role of UniFast-HGR as a practical dependence regularizer for multimodal feature learning. The method avoids covariance decomposition and removes the $K \times K$ covariance storage bottleneck that limits covariance-based HGR/CCA objectives in high-dimensional regimes. Its full objective has $O(m^2K)$ structural computation for local batch size $m$ and feature dimension $K$, while OptFast-HGR reduces the projected structural computation to $O(mKr)$.

The three core components—cosine alignment, diagonal removal, and variance control—address the main computational and numerical bottlenecks of HGR-style estimation. They allow the objective to scale to high-dimensional features, such as $K = 1024$ in vision transformers, while keeping stable behavior under noise, modality imbalance, and spurious correlations. The local-batch formulation also fits standard data-parallel training, since the Gram operator is formed on the per-device feature block.

Overall, UniFast-HGR and OptFast-HGR improve computational efficiency while maintaining competitive performance in image classification, remote sensing, retrieval, and multimodal emotion recognition. The methods are suitable for scalable multimodal feature extraction under both large-scale and resource-constrained settings.

