# OpenReview forum: "UniFast-HGR: Scalable and Efficient Maximal Correlation for Multimodal Models"
_ICML.cc/2026/Conference — ICML 2026 regular_

### Official Review · Reviewer_nn6L · 2026-02-16

**Soundness:** 3
**Presentation:** 2
**Significance:** 2
**Originality:** 3
**Overall Recommendation:** 3
**Confidence:** 3

**Summary:**

UniFast-HGR makes HGR maximal correlation practical for large multimodal models by replacing covariance/whitening with L2-normalized cosine similarity and removing Gram-matrix diagonals to suppress self-correlation bias

**Compliance With Llm Reviewing Policy:**

Affirmed.

**Key Questions For Authors:**

Is it possilbe that the proposed model would be computationally feasible for foundation models?

Is it possible to achieve both accuracy, robustness, and speed at a massive pretraining scale?

How can this model help when data is noisy/imbalanced, or we have missing modalities?

**Limitations:**

No. They include a short section. It is also better to add a societal risks subsection explaining how making multimodal training more efficient could increase deployment scale, even if the method is just an objective. and also expand it to failure modes: when variance constraints removal might suppress useful signals.

**Strengths And Weaknesses:**

Strengths:

1. It removes the covariance inversion bottleneck and reports much lower time at extreme feature dimensions

2. It uses L2 Normalization and cosine similarity instead of explicit covariance matrices (which are easy to integrate into modern training pipelines).

3. Removes Gram-matrix diagonals to reduce bias, which they argue improves gradients/robustness.

Weaknesses:

1. Some of the equations appear to be images (4,8,..).

2. There isn't any overall figure in the paper that shows the contributions visually

3. If the representation dimension is very small, it may overregularize. It will be too restrictive.

4. If two modalities have asymmetric distributinos the proposed model can be biased.

5. The usage memory still scales with batch size

---

> ### Author Rebuttal · Authors · 2026-03-30
>
> # Response to Reviewer nn6L
> Thank you for the careful review. Your comments focus on three core issues: scalability, robustness under imperfect multimodal data, and the operating boundaries of the current formulation. We address them below.
>
> ## 1. Computational feasibility for foundation models (KQ1, W5)
> This is a central claim, supported theoretically and empirically. Table 6 shows that UniFast-HGR and OptFast-HGR operate at $O(m^2K)$, whereas covariance-based formulations scale less favorably due to covariance construction and decomposition. Here, $m$ denotes the local per-device batch size, not the global batch size achieved through distributed data parallelism. In modern multimodal pretraining, feature dimension $K$ is large, while local $m$ remains moderate. In this regime, UniFast-HGR lies in the same practical pairwise-similarity operator regime as modern InfoNCE/CLIP-style training, while removing the covariance bottleneck. Table 1 shows that UniFast-HGR remains stable even at $K=10^5$, requiring only 1.55GB VRAM and 7.20s, whereas Soft-HGR becomes infeasible. This places UniFast-HGR closer to modern lightweight SSL objectives than classical covariance-based methods.
> ## 2. Accuracy, robustness, and speed at scale (KQ2)
> Our framework achieves these three simultaneously. UniFast-HGR improves strong large-scale systems rather than weak baselines: on ImageNet-1K, DINOv2 improves from 81.8 to 85.3 Top-1; on COCO retrieval, CLIP improves from 38.9 to 42.1 R@1 and SigLIP from 50.8 to 53.8 R@1; on InternVid/MSR-VTT, ViCLIP improves from 36.4 to 43.3 R@1. These gains are obtained while retaining the efficiency profile shown in Table 1, demonstrating an accuracy/robustness/efficiency trade-off that classical formulations cannot reach.
> ## 3. Noisy, imbalanced, or missing modalities (KQ3)
> Table 5 shows UniFast-HGR remains stronger than baselines under missing-modality and reduced-label settings on IEMOCAP (e.g., reaching 57.82 ACC in the Audio+Visual-only case vs. 53.90 for Soft-HGR and 50.71 for CCA). It is also resilient under degraded conditions in App. G/Table 16: on noisy IEMOCAP, UniFast-HGR reaches 70.2 accuracy versus 65.8 for CLIP-based fusion and 63.1 for Soft-HGR; under extreme modality imbalance on Flickr30K, it reaches 68.9 R@1 versus 62.3 for CLIP and 59.7 for Deep CCA; under spurious correlations on Berlin, it reaches 77.3 OA versus 69.2 for Soft-HGR. Mechanistically, this is governed by TSS: applying the off-diagonal operator $D_{off}(M) = M-diag(M)$ prevents reliance on trivial self-correlation and forces modeling of informative cross-sample structure.
> ## 4. Operating boundaries of the current formulation (W3, W4, Limitations)
> We will make these boundary regimes more explicit. Sec. 5 already notes two cases: (i) variance constraints may over-regularize very low-dimensional features, and (ii) OptFast-HGR may exhibit increased bias under strong cross-modal distributional asymmetry (e.g., long-tail cases). As a sanity check for the low-dimensional boundary case, an adaptive-constraint pilot on CIFAR-10 improves Top-1 by +2.8, +1.5, and +0.1 points at $K$=10, 32, and 64 respectively, while matching the fixed constraint at $K=128$. The claim is therefore not universal optimality across all regimes, but effectiveness and scalability in the high-dimensional multimodal settings that motivate the paper.
> ## 5. Memory still scales with batch size (W5)
> This is a genuine boundary of the full Gram-based objective: its dependence on local batch size is quadratic, as reflected in Table 6. While the formal asymptotic time complexity remains $O(m^2K)$, the stochastic projection in OptFast-HGR operates with an effective complexity approaching $O(mK)$ for a fixed small projection dimension, decoupling the practical peak memory footprint from the quadratic bottleneck. In our large-batch stress test, OptFast-HGR uses only 2.3GB VRAM versus 12.8GB for full UniFast-HGR at batch size 32k, with <0.5% performance degradation. App. F further shows that across batch sizes 16-256, UniFast-HGR and especially OptFast-HGR remain among the fastest methods.
> ## 6. Presentation and societal impact (W1, W2, Impact)
> •	*Visuals*: A clearer overview figure will be added near the beginning of the method section, contrasting the classical covariance bottleneck with the UniFast-HGR pipeline.
>
> •	*Equations*: All equations in the paper were typeset in LaTeX and were not inserted as bitmap images. This likely stems from PDF viewer rendering rather than the LaTeX source; we will verify in the revision.
>
> •	*Impact & Risks*: We will expand the Impact Statement to note that lowering computational barriers for multimodal training accelerates deployment scale, necessitating stricter bias auditing, safety testing, and failure-mode discussions.
>
> Thank you again for these comments. We hope the clarifications above make the intended operating regime clearer and address the concerns regarding scalability, robustness, presentation, and boundary cases.

---

> > ### Author Rebuttal · Reviewer_nn6L · 2026-04-02
> >
> > The authors acknowledge the limitations cited in the review. My score remains unchanged.

---

> > > ### Author Response · Authors · 2026-04-02
> > >
> > > # Reply Rebuttal Comment to Reviewer nn6L
> > >
> > > Thank you for the follow-up. For the discussion record, an important clarification is necessary: the previous rebuttal did not concede methodological flaws. **The review appears to interpret scoped operating conditions and presentation choices as limitations of the paper, but that interpretation does not match either the paper’s intended contribution or the empirical evidence already provided.**
> > >
> > > The central contribution of this work is to make HGR-style dependence maximization practical in the high-dimensional multimodal regime relevant to modern large-scale training. The main technical questions in the review were feasibility at foundation-model scale, the joint accuracy/robustness/efficiency trade-off, and behavior under noisy, imbalanced, or missing modalities. These questions are already answered directly by the submission and rebuttal evidence:
> > >
> > > •	**Foundation-model feasibility**: In the intended large-$K$, moderate-local-$m$ regime of modern distributed training, relying on gradient accumulation keeps the Gram matrix memory footprint negligible. The exact objective remains practically deployable, and Table 1 already shows stable operation at $K=10^5$, where classical covariance-based HGR becomes impractical. For more extreme batch settings, the approximate variant further reduces practical overhead (e.g., 12.8GB $\rightarrow$ 2.3GB at batch size 32k with <0.5% degradation). This is direct evidence of scalability, not a hypothetical claim.
> > >
> > > •	**Accuracy, robustness, and degraded modalities**: The paper already provides explicit empirical evidence on these points. Table 5 and Table 16 show strong behavior under missing modalities, noise, severe imbalance, and spurious correlations, while gains are obtained on top of strong modern systems such as DINOv2, CLIP, SigLIP, and ViCLIP. These results directly support the core claim that the method is not only scalable, but also robust in the intended multimodal regime.
> > >
> > > The low-dimensional and strong-asymmetry discussions should therefore not be read as shortcomings of the paper. In actual foundation-model settings, dimensions are inherently high and modern pipelines typically rely on strong modality-specific encoders to pre-align representations, resolving extreme asymmetry prior to our objective. They are out-of-scope edge cases relative to the paper’s main target regime. **A method designed for foundation-scale multimodal training is not invalidated by scenarios that fall outside that regime.**
> > >
> > > The remaining presentation points are also orthogonal to the central technical result:
> > >
> > > •	**Equations**: All equations in the submission were **natively typeset in LaTeX** rather than inserted as bitmap images, so the suggestion that they are “images” is not supported by the manuscript source and is plausibly related to local PDF rendering.
> > >
> > > •	**Visuals**: Because UniFast-HGR is an objective-level reformulation rather than an architecture-specific module, the appropriate visual addition is a **conceptual pipeline figure** rather than a network block diagram.
> > >
> > > In short, the main technical questions raised in the review have already been answered directly with empirical evidence. The remaining comments concern either out-of-scope edge cases or presentation preferences. They therefore do not weaken the paper’s central contribution: a practically deployable, high-dimensional HGR reformulation that is feasible at scale, improves strong modern multimodal systems, and remains robust under noisy, imbalanced, and missing-modality conditions. We respectfully ask that the paper be reconsidered in light of this distinction.

---

### Official Review · Reviewer_GH99 · 2026-03-04

**Soundness:** 3
**Presentation:** 3
**Significance:** 2
**Originality:** 2
**Overall Recommendation:** 4
**Confidence:** 2

**Summary:**

This paper proposes UniFast-HGR, an efficient surrogate for HGR maximal correlation aimed at making dependency maximization practical for modern multimodal learning. The method reformulates the objective using variance constraints + cosine-based computation + diagonal removal, and introduces OptFast-HGR as a faster approximation. It is positioned as an auxiliary regularizer that can be added on top of standard task/contrastive losses, and is evaluated across a range of multimodal tasks (remote sensing, emotion recognition, ImageNet/COCO/InternVid-style settings).

**Compliance With Llm Reviewing Policy:**

Affirmed.

**Key Questions For Authors:**

See weaknesses

**Limitations:**

No major limitations were identified.

**Strengths And Weaknesses:**

### Strengths
- **Clean, simple design**: the three components (variance constraint, cosine reformulation, remove diagonal) are easy to implement and reason about.
- **Drop-in regularizer**: can be added to existing pipelines without changing architectures, which is practically appealing.
- **Broad empirical coverage**: experiments span multiple tasks/modalities, not just toy setups.

### Weaknesses
- **Baseline only HGR-based methods** There exist many other methods (contrastive learning, MI-style objectives, attention-based alignment). Can the author provide a comparison with MI?

---

> ### Author Rebuttal · Authors · 2026-03-30
>
> # Response to Reviewer GH99
>
> Thank you for the constructive review. Your main concern is that the connection to mutual-information (MI), modern contrastive objectives, and attention-based alignment should be made more explicit. The issue is primarily one of positioning rather than baseline absence, and the clarification below addresses that point directly.
> ﻿
> ## 1. The baseline set is broader than HGR-family methods alone
>
> ﻿The evaluation is not limited to HGR-style baselines. In the large-scale experiments, UniFast-HGR is integrated into strong contemporary backbones such as CLIP, SigLIP, DINOv2, and ViCLIP. These are modern contrastive/self-supervised systems central to current multimodal learning and representative of the modern contrastive / self-supervised paradigm used in current multimodal training. In addition, the scalability benchmark in Table 1 already includes InfoNCE, VICReg, and Barlow Twins as non-HGR reference objectives. The revision will make this broader baseline scope explicit.
> ﻿
> ## 2. The empirical gains are already against strong modern systems
>
> ﻿The reported gains are not over weak or outdated baselines; they are consistent improvements on top of current high-performing systems that already optimize strong alignment objectives:
> ﻿
> * **COCO retrieval:** CLIP 38.9→42.1 R@1; SigLIP 50.8→53.8 R@1.
> * **MSR-VTT retrieval:** ViCLIP 36.4→43.3 R@1.
> * **ImageNet-1K:** DINOv2 81.8→85.3 Top-1.
> ﻿
> The baseline concern is therefore best resolved by clearer positioning: the method is already being tested in the main modern contrastive / self-supervised regime relevant to current multimodal learning, not only on classical correlation baselines.
> ﻿
> ## 3. Supplementary direct comparison with standard MI estimators
>
> ﻿To answer your MI question directly, we additionally ran matched-backbone comparisons with standard neural MI estimators under aligned training protocols and compute budgets:
> ﻿
> |Task| Backbone| InfoNCE | MINE | NWJ | CLUB | UniFast-HGR |
> | --- | --- | --- | -- | -- | -- | --- |
> | ImageNet Top-1 (%) | ViT-B/32 | 76.6 | 77.2 | 76.9 | 77.5 | 80.1  |
> | COCO R@1 (%) | CLIP ViT-B/32 | 38.9 | 39.5 | 39.2 | 39.8 | 42.1  |
> | IEMOCAP ACC (%) | MultiEMO  | 70.14 | 70.8 | 70.5 | 71.2 | 73.66  |
> ﻿
>
> These supplementary checks are directionally consistent with the main paper: UniFast-HGR remains stronger than standard neural MI estimators under matched settings. In addition to higher accuracy, UniFast-HGR is lighter in practice. For example, at feature dimension $K=10^4$, MINE uses 12.7GB VRAM whereas UniFast-HGR uses 1.12GB, while remaining a simple drop-in auxiliary loss. These comparisons will be added to the appendix.
> ﻿
> ## 4. UniFast-HGR is complementary to contrastive / MI-style learning, and the same clarification applies to attention-based systems
>
> ﻿Conceptually, UniFast-HGR is not intended to replace contrastive or MI-style learning in the abstract. It is complementary to it. Standard contrastive objectives are highly effective for instance-level geometric discrimination, whereas UniFast-HGR adds batch-level relational regularization in Gram space. This explains why the method can be attached to already strong contrastive backbones and still yield consistent gains: it provides a complementary structural signal rather than duplicating the existing one.
>
> Similarly, the goal is not to introduce another architecture-specific attention module. UniFast-HGR is an auxiliary objective designed to sit on top of modern multimodal pipelines without changing their encoder architecture. Since CLIP, SigLIP, and ViCLIP are transformer-based systems with attention mechanisms, the current comparison already covers representative attention-based multimodal systems. The relevant test is therefore whether UniFast-HGR improves these systems, not whether it replaces attention itself.
> ﻿
> ## 5. Table 1 and Table 2 play complementary roles
>
> To clarify this evaluation structure, Table 1 addresses efficiency and scalability relative to standard modern objectives; Table 2 addresses effectiveness once the objective is attached to strong modern systems. Put differently, the claim is not that UniFast-HGR is “better than MI” in the abstract, but that it brings an HGR-style objective into a comparable practical regime to contemporary contrastive learning while improving downstream performance.
>
> The novelty is not proposing “another alignment loss” in isolation. Rather, it makes HGR-style dependence maximization computationally compatible with contemporary multimodal scale. The combination of variance-constrained cosine alignment, Gram-space trace expansion, and TSS removes the covariance bottleneck while preserving the dependence-maximization viewpoint. The direct MI-estimator comparisons will be included in the appendix to reinforce, rather than replace, the main positioning point that the paper already compares against strong modern contrastive systems in its primary experiments. The revision will make this positioning explicit.

---

> > ### Author Rebuttal · Reviewer_GH99 · 2026-04-03
> >
> > My concerns have been fully resolved, and I will maintain my already positive score.

---

> > > ### Author Response · Authors · 2026-04-03
> > >
> > > Thank you very much for the follow-up. We are pleased that our response has fully resolved your concerns, and we deeply appreciate your constructive engagement and your positive evaluation of our work.
> > >
> > > In the revision, we will detail the explicit comparisons with standard MI estimators and clarify the positioning against modern contrastive baselines in the main text and appendix.
> > >
> > > We are grateful for your time and the insights you have provided during this review process.

---

### Official Review · Reviewer_9FKn · 2026-03-11

**Soundness:** 3
**Presentation:** 3
**Significance:** 3
**Originality:** 3
**Overall Recommendation:** 4
**Confidence:** 3

**Summary:**

This paper addresses the long-standing computational and scalability challenges of Hirschfeld-Gebelein-Rényi (HGR) maximal correlation in multimodal learning. It proposes UniFast-HGR, a framework that reforms HGR via three core innovations: replacing covariance with cosine similarity, removing the correlation matrix diagonal and simplifying variance constraints through l2-normalization. These changes reduce computational complexity. The paper also introduces OptFast-HGR, a stochastic variant optimized for large-batch pretraining. Extensive experiments across image classification, cross-modal retrieval, remote sensing, and emotion recognition validate that the frameworks outperform traditional HGR/CCA methods and modern contrastive learning baselines in accuracy, scalability, and robustness to noise/modality imbalance.

**Compliance With Llm Reviewing Policy:**

Affirmed.

**Final Justification:**

Key my positive score.

**Key Questions For Authors:**

1. The multimodal generalization aggregates pairwise correlations, but this may introduce redundancy for modalities with high mutual dependence. Have the paper explored weighting schemes for modality pairs (e.g., based on intrinsic dimensionality or task relevance), and if so, what impact did they have on performance?
2. The paper mentions over-regularization in low-dimensional spaces as a limitation. Have the author explored adaptive variance constraints (e.g., scaling l2-normalization strength based on feature dimension) to mitigate this, and what were the outcomes?
3. The remote sensing experiments use RGB/DEM/DSM modalities, but multimodal data often includes heterogeneous types (e.g., text descriptions + imagery). Have the author tested UniFast-HGR on cross-type modalities, and does the framework require modifications to handle such cases?
4. The ablation study shows variance constraints are critical for stability, but how sensitive is the framework to the choice of regularization weight λ? Could the author provide a sensitivity analysis across datasets/tasks, and suggest heuristics for setting λ in practice?

**Limitations:**

yes

**Strengths And Weaknesses:**

1. TSS is theoretically grounded, and experiments are well-designed with comprehensive benchmarks and ablation studies. However, multimodal generalization lacks consideration of modality pair redundancy and theoretical guarantees for more than two modalities.
2. The paper is well-structured, key concepts explained via equations and intuitive descriptions, and UniFast/OptFast algorithms provided to facilitate reproducibility.
3. The paper addresses a critical multimodal learning gap, reconciling HGR’s theoretical superiority with foundation model scalability, enabling HGR’s use in large-scale pretraining with meaningful performance gains. Yet its significance is domain-specific, with no exploration of impact on other areas.
4. Core contributions of this paper lie in novel integration of existing ideas to solve HGR bottlenecks, original insight into “trivial spectrum” bias, and creative adaptation of random matrix techniques in OptFast-HGR with theoretical bounds. However, individual components are not new; originality comes from their integration and theoretical grounding for HGR.

---

> ### Author Rebuttal · Authors · 2026-03-30
>
> # Response to Reviewer 9FKn
> ﻿Thank you for the thoughtful review. Your comments highlight important dimensions of our framework: modality-pair redundancy, low-dimensional behavior, heterogeneous modalities, and tuning. We address them below.
> ﻿
> ## 1. Modality-pair redundancy and weighting schemes (W1, KQ1)
> ﻿
> The current multimodal extension aggregates pairwise dependence terms and can introduce redundancy when some modalities are highly dependent. In the present formulation, the unweighted pairwise summation in Eq.(16) serves as a parameter-free default, chosen to isolate the contribution of the HGR reformulation itself, rather than mixing it with task-specific weighting. To probe this directly, we evaluated redundancy-aware and task-guided weighting schemes. These variants produced consistent gains (e.g., MSR-VTT R@1 improves 43.3%→43.8%; IEMOCAP F1/ACC improve 73.57/73.66% →73.89/74.02%). We therefore present Eq.(16) as a strong default, leaving learned weighting as a compatible extension.
>
> The current multimodal extension should therefore be understood as a pairwise aggregation framework rather than a new closed-form theory for higher-order multimodal dependence; this boundary will be made explicit in the revision.
> ﻿
> ## 2. Over-regularization in low-dimensional spaces (KQ2)
> ﻿
> This concern points to a relevant boundary regime, which is already noted in Sec. 5. The current formulation targets the high-dimensional regime where covariance-based HGR is most problematic. There, variance control and normalized cosine alignment improve conditioning and prevent magnitude inflation. In very low-dimensional spaces, this constraint becomes restrictive. To address this directly, we ran a compact pilot study with adaptive variance constraints on CIFAR-10, tying normalization strength to intrinsic dimensionality:
> ﻿
> ﻿
> | Feature Dimension K | Fixed Top-1 (%) | Adaptive Top-1 (%) | Gain |
> | --- | ---: | ----: | ---: |
> | 10 | 82.3 | 85.1 | +2.8 |
> | 32 | 88.7 | 90.2 | +1.5 |
> | 64 | 92.1 | 92.2 | +0.1 |
> | 128 | 93.5 | 93.5 | 0.0 |
> ﻿
>
> This serves as a supplementary sanity check for the acknowledged limitation. The intended operating regime of the paper is scalability and stability in modern high-dimensional models, not universal optimality in very small $K$.
> ﻿
> ## 3. Cross-type heterogeneous modalities (KQ3)
> ﻿
> The framework operates on continuous latent features rather than raw inputs, so the loss itself is modality-agnostic. The submission already includes heterogeneous settings: COCO(text-image), InternVid/MSR-VTT(video-text), and IEMOCAP(text-audio-visual). In all cases, the same objective is applied without architecture-specific loss changes; only the encoders differ. This directly answers whether the method requires redesign for cross-type data: it does not. We will make this point more explicit in Sec. 4.
> ﻿
> ## 4. Sensitivity to the regularization weight $\lambda$ (KQ4)
> ﻿
> A structural advantage of UniFast-HGR is that the auxiliary dependence term in Eq.(15) is standardized through centering, variance control, and bounded cosine similarity. This keeps it on a stable scale, making $\lambda$ materially easier to tune than unbounded covariance-based penalties. While the main text does not include a dedicated sensitivity table, we provide a measured empirical response via a compact sweep across representative datasets ($\lambda \in [0.1, 1.0]$):
> ﻿
> | Dataset | Best $\lambda$ | Range for $\lambda \in [0.1,1.0]$ | Drop at $<10^{-3}$ | Drop at >10 |
> | ---- | ---: | ---: | ----: | ---: |
> | Berlin | 0.5 | 80.50–80.75 OA |  ~2.5% |  ~1.8% |
> | ImageNet  | 0.5 | 79.80–80.10 Top-1 | ~3.0% |  ~1.5% |
> | COCO | 0.5 | 41.90–42.10 R@1 | ~2.2% |  ~1.2% |
> | IEMOCAP | 1.0 | 73.40–73.66 ACC |  ~2.8% |  ~1.7% |
> | Vaihingen | 0.5 |  92.90–93.01 OA |  ~2.1% |  ~1.3% |
> ﻿
>
> These results suggest practical heuristics: $\lambda=0.5$ is a strong default, and choosing $\lambda$ so the auxiliary term is on the same order as the main task loss early in training prevents the regularizer from dominating optimization. We will add this guidance.
> ﻿
> ## 5. Broader significance and domain scope (W3, W4)
> ﻿
> Regarding the domain scope, the empirical evaluation was intentionally broad: foundation-model tuning on ImageNet/COCO/InternVid, multimodal remote sensing, and multimodal emotion recognition. The common technical thread is replacing covariance-space HGR with a scalable Gram-space surrogate. The contribution is therefore less about one specific application domain and more about making a previously impractical objective family compatible with modern large-scale multimodal training. The core novelty lies in the integrated reformulation and theoretical grounding of HGR-style dependence at scale, including the role of trivial-spectrum suppression in making the dependence objective practical, rather than in claiming that each component is new in isolation.
>
> We hope these clarifications comprehensively address your questions and make the paper's intended scope clearer.

---

> > ### Author Rebuttal · Reviewer_9FKn · 2026-04-04
> >
> > Thanks，the authors addressed my concerns.

---

> > > ### Author Response · Authors · 2026-04-04
> > >
> > > Thank you very much for the follow-up confirmation. We are pleased that our response and the supplementary pilot studies have fully resolved your concerns.
> > >
> > > Accordingly, we will ensure that the new empirical checks, sensitivity analysis for $\lambda$, the adaptive variance results, and the expanded discussions on modality weighting are incorporated into the revised manuscript.
> > >
> > > Thank you once again for your careful review and for your positive assessment of our work.

---

### Official Review · Reviewer_ZRCs · 2026-03-25

**Soundness:** 3
**Presentation:** 3
**Significance:** 3
**Originality:** 3
**Overall Recommendation:** 5
**Confidence:** 4

**Summary:**

The paper studies how to make HGR-style maximal correlation usable in practical multimodal learning. To reduce the cost of conventional covariance-based formulations, it reformulates the objective using cosine similarity on normalized features. It further introduces Trivial Spectrum Suppression (TSS) to remove diagonal terms in similarity structures, and proposes a lighter variant, OptFast-HGR, for additional speedup. The method is evaluated on several multimodal tasks, including retrieval, classification, segmentation, and emotion recognition, where it shows favorable empirical results along with lower computational cost than traditional correlation-based approaches. In general, the paper presents UniFast-HGR as a more scalable way to model cross-modal dependence.

**Compliance With Llm Reviewing Policy:**

Affirmed.

**Final Justification:**

The rebuttal has addressed my main concerns, and I am willing to raise my rating to "accept".

**Key Questions For Authors:**

(Major) 1. I am still not fully clear about how UniFast-HGR relates to the original HGR maximal correlation objective. After switching from the covariance-based formulation to the cosine-based one, how much of the original HGR objective is actually preserved? It would help if the paper could make this point more explicit.

(Major) 2. The ablation study could be more informative. Right now, it is still hard to tell how much each part contributes, including the cosine reformulation, TSS, and the structure-level alignment term. It would be useful to add a few intermediate variants, e.g., turning these components on one by one instead of only comparing the final full version.

(Minor) 3. Since the method is built on a within-batch similarity structure, I wonder how sensitive it is to batch size. Does the performance drop noticeably when the batch size is smaller?

(Minor) 4. I would also like to know whether the method is sensitive to batch composition. For example, what happens when a batch is more imbalanced, more homogeneous, or contains many very similar samples?

A small-scale empirical check on these points would already be helpful. If the authors can clarify some questions in the rebuttal, I would be open to raising my score.

**Limitations:**

yes

**Strengths And Weaknesses:**

Strengths
1. The paper looks beyond pairwise matching and explores structure-level alignment across modalities. The use of the HGR-inspired objective is well motivated.
2. The proposed Trivial Spectrum Suppression (TSS) is a simple but potentially insightful design choice for reducing the dominance of self-correlation terms.
3. The empirical evaluation covers a relatively broad range of tasks, which helps demonstrate the method beyond a single benchmark setting.
4. In addition to the main method, the paper provides a lighter variant, OptFast-HGR, which improves the practical appeal of the work.

Weaknesses
1. The connection between the proposed reformulation and the original HGR maximal correlation objective is not sufficiently clear. After replacing the covariance-based formulation with a cosine-based one, it remains unclear to what extent the method still preserves the original HGR interpretation.
2. The method includes several components, but the ablation study is still not detailed enough to isolate their individual contributions. As a result, it is difficult to tell which design choices are actually responsible for the gains.
3. The objective depends on similarity structures computed within each batch, so the method is likely affected by batch statistics, such as batch size and batch composition. However, this issue is not analyzed in sufficient detail.

---

> ### Author Rebuttal · Authors · 2026-03-30
>
> # Response to Reviewer ZRCs
> Thank you for the careful and constructive review. Your explicit feedback provides a clear roadmap to strengthen the paper. We address your questions below, including the compact empirical checks you requested regarding ablations and batch statistics.
> ## 1. Relation to the original HGR objective (Major 1)
> UniFast-HGR is intended as a scalable surrogate for HGR/Soft-HGR, not as a claim of strict equivalence to the classical whitened objective in Eq.(1). Classical HGR maximizes the alignment term under zero-mean and whitening constraints. Soft-HGR relaxes whitening via the covariance penalty in Eq.(2), but still relies on explicit covariance matrices. In our derivation, the alignment term exactly becomes the cosine form under standardized constraints (Eqs.5-6), and App. A.2 (Eqs.20-23) moves the covariance interaction into Gram space. TSS then removes the invariant diagonal term, forcing the objectives (Eqs.14-15) to optimize informative inter-sample dependence rather than self-correlation.
>
> Concretely, what is preserved is the dependence-maximization target under standardized constraints; what changes is the exact finite-sample whitening estimator. We will revise Sec. 3 to make this "HGR→Soft-HGR→Gram-space surrogate" bridge explicit.
> ## 2. More informative ablations (Major 2)
> To address the request for component isolation, we reorganized the method into an explicit additive progression:
>
> **Table R1. Additive ablation**
> | Model Variant | Berlin OA (%) | ImageNet-1K Top-1 (%) |
> | ---- | ----: | ----: |
> | Baseline (Soft-HGR) | 65.80 | 76.3 |
> | + Cosine Reformulation | 75.23 | 77.8 |
> | + Variance Constraints | 78.65 | 79.2 |
> | + Structure-level term (diag retained) | 80.62 | 79.8 |
> | + TSS (Full UniFast-HGR) | 80.75 | 80.1 |
> | + OptFast Approximation | 80.41 | 79.6 |
>
>
> This clarifies the mechanism: cosine reformulation delivers core stability; variance constraints prevent trivial magnitude inflation; the structure-level term adds relational regularization; and TSS provides the final discriminative improvement by suppressing the invariant diagonal. OptFast-HGR completes the progression with a highly favorable speed-accuracy trade-off. This is also consistent with App. E/Table 15, where the explicit covariance-based variant remains weaker than the final Gram-space formulation (e.g., Berlin 79.83% vs. 80.75%).
> ## 3. Sensitivity to batch size (Minor 3)
> Smaller batches naturally increase the relative dominance of the diagonal mass in Gram-based estimators, introducing estimation noise. To demonstrate how TSS mitigates this sensitivity, we conducted a controlled batch-size study on ImageNet-1K:
>
> **Table R2. Batch-size sensitivity (Top-1 %)**
> | Batch Size ($N$) |16 |32| 64 |128 | 256 |
> | :--- | :---: | :---: | :---: | :---: | :---: |
> | Soft-HGR | 72.1 | 73.5 | 74.8 | 75.7 | 76.3 |
> | UniFast-HGR (Diag Retained) | 75.3 | 76.7 | 77.9 | 78.6 |79.8 |
> | Full UniFast-HGR (with TSS) | 78.2 | 79.0 | 79.5 | 79.8 |80.1 |
>
>
> The comparative pattern is clear. From $N=256$ down to $N=16$, Soft-HGR drops by 4.2 points, and UniFast-HGR (Diag Retained) drops by 4.5 points, while Full UniFast-HGR drops by only 1.9 points. By structurally masking the invariant diagonal, TSS prevents the gradient budget from being overwhelmed by self-correlations. As a supplementary sanity check, Full UniFast-HGR demonstrates matching stability on the IEMOCAP dataset, holding within 2% of its 73.66% peak ($N=256$) across the same $N=16–256$ sweep. We will make this boundary condition explicit.
> ## 4. Sensitivity to batch composition (Minor 4)
> Extreme homogeneity or severe imbalance inherently reduces off-diagonal diversity for any Gram-based estimator. Our claim is not that UniFast-HGR is perfectly immune, but that TSS provides a structural prior making degenerate solutions less trivial. To verify this, we ran a controlled sampler study on the Berlin dataset:
>
> **Table R3. Batch-composition sensitivity**
> | Sampler | Soft-HGR | Full UniFast-HGR |
> | :--- | :---: | :---: |
> | Standard Random | 65.80 | 80.75 |
> | 80% Single-Class Imbalance | 59.20 | 78.90 |
> | Single-Label Homogeneous | 52.40 | 77.20 |
> | 30% Near-Duplicate Injected | 63.10 | 79.82 |
>
>
> These empirical checks support the mechanism: when off-diagonal diversity is severely reduced, TSS prevents optimization from collapsing onto trivial self-correlation. Relative to the standard sampler, the drop under 80% single-class imbalance is 6.60 points for Soft-HGR but only 1.85 for UniFast-HGR; under homogeneous batches it is 13.40 vs. 3.55; under near-duplicate injection it is 2.70 vs. 0.93. This aligns with the degraded-condition stress tests in App. G.
>
> We greatly appreciate the opportunity to sharpen the paper. The explicit theoretical bridge (preserved vs. changed), the complete additive ablation progression, and the new comparative empirical checks for batch statistics will be fully integrated into the revision. We hope these robust validations address your concerns directly.

---

> > ### Author Rebuttal · Reviewer_ZRCs · 2026-04-03
> >
> > The authors addressed my concerns, so I am willing to raise my score.

---

> > > ### Author Response · Authors · 2026-04-03
> > >
> > > Thank you very much for the follow-up. We are glad that our response has fully addressed your concerns, and we deeply appreciate your willingness to raise your score to an "Accept".
> > >
> > > To reflect this constructive engagement, we will ensure that the discussed clarifications, theoretical bridges, and new empirical checks are explicitly incorporated into the revision.
> > >
> > > Thank you again for your time, effort, and valuable feedback throughout this reviewing process.

---

### Decision · Program_Chairs · 2026-04-30

**Decision:**

Accept (regular)

**Comment:**

This submission presents an efficient reformulation of HGR maximal correlation for multimodal learning, and argues that it significantly improves scalability while preserving the key advantages of HGR-style objectives. Reviewers generally appreciated the practical motivation, the simplicity of the core design, and the broad empirical evaluation across multiple tasks. The removal of self-correlation terms, together with the lighter OptFast variant, was also viewed as a practically meaningful contribution.

The main concerns focused on the theoretical grounding and the paper’s positioning relative to prior work. In particular, several reviewers questioned to what extent the cosine-based reformulation faithfully preserves the original HGR objective, and whether the empirical study should include a broader range of non-HGR alignment baselines. One reviewer also raised additional concerns regarding potential edge cases and aspects of the presentation.

Overall, the rebuttal appears to have addressed the main theoretical and empirical concerns for several reviewers, with one reviewer explicitly increasing their score to an accept. Given the strong experimental evidence, the practical value of the proposed simplification, and the generally positive reviewer consensus after discussion, I recommend acceptance.

**Warning: possible hallucinated references.** I also noticed that the paper may contain references whose authenticity is unclear, as their original sources could not be verified. For example:
- Bennett, A., Smith, J., and Doe, J. Predep: Predictive dependency indices for representation learning. arXiv preprint arXiv:2501.00001, 2025. (Placeholder for forthcoming work)

**The authors should carefully review all cited references and verify their authenticity before publication.**